# *Klebsiella* ARO112 promotes microbiota recovery, pathobiont clearance and prevents inflammation in IBD mice

Vitor Cabral [1,6,9] ✉, Rita A. Oliveira [1,7,9] ✉, Margarida B. Correia[1], Miguel F. Pedro[1], Marc García-Garcerá [2], Carles Ubeda [3,4] & Karina B. Xavier [1,5] ✉

Precise microbiota modulation towards improving immune function and metabolic homeostasis is a major goal in clinical research. It is also critical for reducing pathogen invasion or pathobiont expansion, contributors to epidemic Inflammatory Bowel Diseases (IBD), where recurrent antibiotic treatments often exacerbate microbiota imbalances. Within the thousands of strains of a natural gut microbiota, we previously identified a specific *Klebsiella* strain, ARO112, capable of promoting resistance to, and clearance of, pathogenic Enterobacteriaceae. Here, we assess its therapeutic potential using a comprehensive genomic and phenotypic analysis and experiments in mouse models of IBD. We demonstrate that ARO112 not only exhibits a safety profile comparable to the widely used probiotic *Escherichia coli* Nissle 1917, but also has a reduced capacity to acquire antibiotic resistance, via horizontal gene transfer, and to capture iron, thereby bypassing major concerns associated with pathogenic Enterobacteriaceae strains. In antibiotic-treated, genetically predisposed IBD mice, ARO112 accelerates pathobiont clearance, promotes the recovery of microbiota diversity, elevates intestinal butyrate concentration, and prevents mild inflammation. Moreover, even in the absence of pathogen infection, ARO112 prevents severe inflammation-driven pathology in a chemically-induced colitis model. Our findings highlight ARO112 as a potential biotherapeutic agent that disrupts inflammation-treatment-infection cycles characteristic of chronic gut inflammatory diseases.

The gut microbiota plays a crucial role in human health and disease. This abundant and diverse intestinal community of microorganisms performs essential functions, including providing colonization resistance to invasion by potential pathogenic bacteria[1–4]. This protective capacity can be significantly compromised by environmental perturbations that alter the population composition, such as changes in diet or medical treatments, like antibiotics[5–9]. These shifts in microbial populational dynamics can be challenging to rectify, leaving patients more susceptible to infections and other medical complications[8,10].

[1]Gulbenkian Institute for Molecular Medicine, Oeiras, Portugal. [2]Department of Fundamental Microbiology, University of Lausanne, Lausanne, Switzerland. [3]FISABIO, Valencia, Spain. [4]CIBER en Epidemiología y Salud Pública, Madrid, Spain. [5]Faculdade de Medicina da Universidade de Lisboa, Lisbon, Portugal. [6]Present address: Lukasiewicz Research Network PORT, Wroclaw, Poland. [7]Present address: Duchossois Family Institute, The University of Chicago, Chicago, IL, USA. Please cite this article as Cabral & Oliveira et al., 2025. [9]These authors contributed equally: Vitor Cabral, Rita A. Oliveira. ✉e-mail: vitor.cabral@port.lukasiewicz.gov.pl; aaoliveira@uchicago.edu; karina.xavier@gimm.pt

The pursuit for interventions to restore dysbiotic gut microbiotas has recently gained momentum. Currently, both basic research and clinical studies encompass a range of approaches. A common approach relies on fecal microbiota transplants (FMT)[11], where the fecal microbiota of a healthy individual is given to an individual with a disease. Although the United States Food and Drug Administration has recently approved FMT for the treatment of recurrent *Clostridioides difficile* infections in humans[12], the undefined nature of these procedures poses safety risks[13,14]. A more precise approach involves either single-bacterium or multi-strain cocktails. Next-generation probiotic bacteria, defined as bacterial species native to the host or target location that upon ingestion and/or colonization provide benefits, offer promise for restoring microbiota functions[15]. This requires an extensive understanding of the benefits, potential drawbacks, and the organism's safety profile, all of which are significant to its therapeutic importance.

Patients with high susceptibility to infections often undergo extensive antibiotic treatments, rendering them more susceptible to recurrent infections[16,17]. These patients require innovative approaches to complement current treatments and break the cycle of disease recurrence. Inflammatory Bowel Diseases (IBD) are among such diseases where extensive antibiotic treatments can render the patient more susceptible to frequent infections and recurrence of acute disease symptoms[18,19]. IBD are a group of multi-factorial pathologies with a complex etiology, where genetic predisposition, gut microbiota, and environmental conditions play a role[20–23]. Three outcomes emerge as key aspects of disease cycles that afflict IBD patients: gut microbiota imbalances[24], increased and persistent intestinal inflammation[25], and susceptibility to intestinal infections[26]. To manage inflammation flares, IBD patients are often prescribed antibiotics or anti-inflammatory medication[27]. However, these treatment options can inadvertently harm the microbiota, often exacerbating the proinflammatory intestinal environment[16], thus perpetuating the cycle. The consequent loss of natural protection provided by a balanced microbiota often leads to infections by Enterobacteriaceae bacteria, like the adherent-invasive *Escherichia coli* (AIEC), which can also trigger inflammatory episodes[28].

Probiotics are often considered as potential alternative or complementary treatments for intestinal diseases, but their benefits for complex pathologies, such as IBD have been limited[29]. The probiotic strain *E. coli* Nissle 1917 (EcN) has been proposed as a potential therapy for IBD[30]. However, despite various proposals and even clinical trials[31], the European Society for Clinical Nutrition and Metabolism, recommends against probiotics for treatment or prevention of Crohn's Disease[32], owing to lack of promising results obtained thus far. Therefore, therapeutic probiotics warrant further research, as their efficacy in patients has been inconsistent[33–36].

Recent studies by our laboratory and others have added encouraging evidence that non-*pneumoniae Klebsiella* microbiota species and strains can contribute significantly to protection against Enterobacteriaceae colonization[37–41]. We previously identified and validated nutrient competition as the main driver of colonization resistance and displacement by *Klebsiella* sp. ARO112 against *E. coli* K-12 MG1655. We further showed that ARO112 staved off *Salmonella enterica* Typhimurium's gut expansion and prolonged host survival after infection[37]. Following work by others identified *Klebsiella oxytoca* strains present in children's microbiota that protect against invading *Klebsiella pneumoniae*[38], as well as against *S.* Typhimurium[40], while human isolates of were found to be associated with protection against enterobacterial pathogens in cancer patients receiving allogeneic hematopoietic stem cell transplantation[39]. These studies support the relevance of these bacteria as protective gut microbiota members both in mice and humans.

Here, we compiled a comprehensive panel of assays to assess the safety of the potential probiotic strain ARO112 by evaluating its pathogenic potential for an assembly of traits commonly associated with Enterobacteriaceae clinical isolates, and we used EcN as a gold standard probiotic for comparison. Our results show that ARO112, compared to other Enterobacteriaceae, has low pathogenic potential, including limited efficiency in acquiring or maintaining plasmids from other bacteria carrying antibiotic resistances. Simultaneously, ARO112 displayed a competitive advantage for colonization against different species from the Pseudomonadota phylum (formerly known as Proteobacteria). More importantly, here we evaluated the protective properties of ARO112 in an IBD pre-clinical mouse model. Our results showed that ARO112 exhibited strong potential as a next-generation probiotic, by promoting butyrate-producing microbiota recovery from antibiotic-induced dysbiosis, preventing mild intestinal inflammation, and accelerating clearance of AIEC, a common pathobiont in IBD patients[42]. Furthermore, when we induced colitis in mice with dextran sulfate sodium (DSS), a chemical agent that causes epithelial injury and is commonly used as a model for chronic inflammation, ARO112 treatment was able to attenuate disease activity, protecting from severe diarrhea and inflammation, even in the absence of AIEC infection. Overall, our potential probiotic ARO112 displays the ability to ameliorate intestinal inflammatory symptoms and pathogen clearance independently, presenting itself as a broad therapeutic possibility that prompts further testing.

## Results

### ARO112 encodes fewer predicted traits associated with nosocomial infections when compared with other Enterobacteriaceae

To address the safety of ARO112 to be used as a probiotic, we started by performing a genome analysis with fifteen Enterobacteriaceae strains to determine the presence of genomic traits generally associated with hospital infections in these bacteria. This analysis comprised five *E. coli* strains, five *Klebsiella pneumoniae* strains, and five non-*pneumoniae Klebsiella* strains, including ARO112 (Fig. 1a, Supplementary Data 1). These strains were selected to incorporate human (clinical) and murine bacterial isolates, including pathogenic, commensal, type or laboratory strains, including an established probiotic strain (*E. coli* Nissle 1917) to be used as a standard of safeness. With this broad strain selection, we expected to include a wide range of encoded virulence traits and bacteria with different pathogenic potential.

A whole genome-based phylogenetic tree segregates the fifteen strains into three major clades, differentiating among *E. coli*, *K. pneumoniae*, and non-*pneumoniae Klebsiella* strains (Fig. 1a). The non-*pneumoniae Klebsiella* clade shows ARO112 clustering with *Klebsiella grimontii* type strain and close to *K. michiganensis* type strain, and *Klebsiella* MBC022 mouse commensal strain clustering with the *K. oxytoca* DSM5175 type strain. This placement of ARO112 closer to *K. grimontii* than *K. michiganensis* was also recently shown by others[38,39], where phylogenetic analysis with new *Klebsiella* isolates placed ARO112 strain close, but not part of, the cluster containing three *K. grimontii* strains[38] meaning that it should no longer be considered a *K. michiganensis*, as initially classified[37]. In the *E. coli* clade, the probiotic EcN and the pathobiont AIEC (LF82) strains were phylogenetically closer to each other than to both non-pathogenic laboratory strains *E. coli* K-12 (MG1655) and *E. coli* B, and to the clinical isolate *E. coli* (Ec1898) (Fig. 1a). In the *K. pneumoniae* clade, the three clinical isolates cluster together (MH258, Kp1012, and Kp834), while the other two *K. pneumoniae* strains (Kpn NCTC9633 and ATCC43816) form a distinct cluster within the clade.

To predict the genome-annotated pathogenic potential of these strains we used 9 available databases, reported in Bacterial and Viral Bioinformatics Resource Center[43] (BV-BRC), for virulence factors, drug targets and transporters, and antibiotic resistance genes. The presence/absence of the pathogenic properties segregated the three clades represented in the phylogenetic tree (Fig. 1b, c; Supplementary Data 2). Interestingly, when comparing the ubiquitous gene products and pathogenic properties per clade (shared by all five strains within a

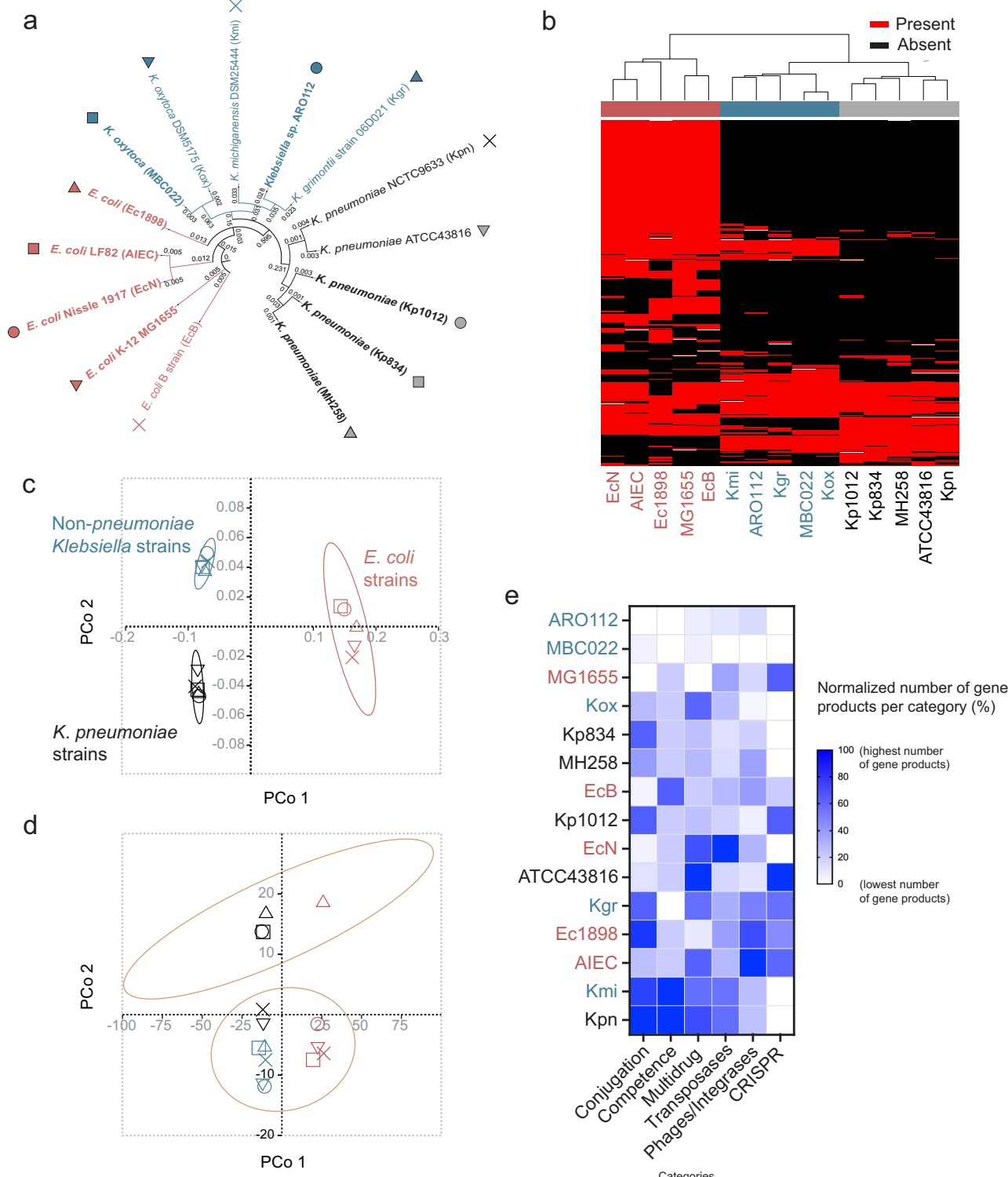

**Fig. 1 | ARO112 genomic comparison with other Enterobacteriaceae strains from the *Klebsiella* and *Escherichia* genera for their pathogenic potential.** **a** Phylogenetic tree based on whole-genome analysis of 15 strains (5 *E. coli* strains, 5 *K. pneumoniae* strains, and 5 non-*pneumoniae Klebsiella* strains). Strains in bold were tested experimentally; detailed description of the different strains in Supplementary Data 1. **b** Heatmap, **c** Principal Coordinate Analysis representing presence/absence of pathogenic properties. **d** Principal

Coordinate Analysis of the different strains using the abundance of pathogenic properties. **e** Normalized quantification of gene products encoding for genes related to clinically-relevant categories (conjugation and conjugal proteins, natural competence, multidrug resistance, transposases, bacteriophages, and integrases, CRISPR systems). **c,d** ellipses represent 90% confidence. **e** highest number per category is set to 100%, lowest number to 0%, remaining numbers are normalized in between these numbers.

clade), there is a higher percentage of gene products that are not ubiquitous to any of the three clades (>75%), compared to the lower percentage of predicted pathogenic traits (<20%) (Supplementary Fig. 1a, Supplementary Data 3).

Upon adding data for the abundance of each predicted pathogenic trait (Supplementary Data 4), instead of the binary analysis of presence/absence, the strains separated in two big clusters along PCo 2 (Fig. 1d), whereas within each cluster the strains still segregate by taxonomy. One cluster included the known multi-drug resistant (MDR) *K. pneumoniae* strain MH258 and the three strains isolated from hospitalized patients (Kp1012, Kp834, Ec1898). We checked the top 10 pathogenic traits driving the separation of the two clusters and found that they include several antimicrobial resistance-associated properties (6/10; Supplementary Data 5). Accordingly, these properties were enriched in the cluster that included the MDR strain MH258. ARO112 was in the second cluster, together with MBC022 that was isolated from non-antibiotic-treated murine gut microbiome, and together with known non-MDR strains like MG1655 and the probiotic EcN (Fig. 1d).

To assess the presence of additional gene products that might have clinical relevance, we analyzed the genomes of each strain for the number of gene products related to the following categories: conjugation, natural competence, multidrug resistance, transposases, phages/integrases, and CRISPR systems. In this analysis, ARO112 consistently shows a lower number of gene products within these categories in comparison with all the other *Klebsiella pneumoniae* and most non-*pneumoniae Klebsiella* strains (Fig. 1e). Importantly, ARO112 was the only *Klebsiella* strain with no annotated gene products related to conjugation or conjugal proteins, which are important for bacteria to acquire antibiotic resistance genes (Fig. 1e). Moreover, even though the genome analysis displays *K. grimontii* as the closest tested relative to ARO112 (Fig. 1a) and the predicted pathogenic traits analysis hints at *K. michiganensis* as closely related to ARO112 (Fig. 1b), both *K. grimontii* and *K. michiganensis* type strains clearly differ from ARO112 regarding gene products for these relevant categories (Fig. 1e).

Overall, these results show that while the predicted pathogenic properties encoded by the bacteria tested align with their taxonomy, the most significant factor distinguishing these bacteria was the abundance of drug resistance genes, which still lightly reflected underlying taxonomic relationships.

## ARO112 shows fewer phenotypic traits associated with nosocomial infections

Given that genome predictions do not always directly translate into phenotypes, we tested a panel of functional in vitro analyses for traits that are generally associated with Enterobacteriaceae pathogenicity in nosocomial infections and gut colonization (Fig. 2a). For that, we selected a subset of the aforementioned strains, including four *E. coli* strains (non-pathogenic K-12 MG1655 strain, pathobiont AIEC, clinical isolate Ec1898, and probiotic EcN to define our baseline for safeness), three *K. pneumoniae* strains (clinical isolates MH258, Kp1012, and Kp834), and two non-*pneumoniae Klebsiella* strains from our strain collection (MBC022, ARO112). We performed these tests under two different laboratory conditions (nutrient rich – Lysogeny Broth, LB – and nutrient poor media – M9 salts minimal medium with glucose) to evaluate if these pathogenic traits vary according to the nutritional environment.

We started by testing resistance to commonly used laboratory antibiotics, as antibiotic resistance is the main driver of clinical concerns associated with Enterobacteriaceae[44], due to their efficient ability to acquire and share antibiotic resistance genes from other bacteria through mechanisms like HGT[45]. Moreover, predicted abundance of antibiotic resistance genes seems to be a major driver in segregating our set of Enterobacteriaceae strains (Fig. 1d). Our results showed that ARO112, like MG1655, EcN, and MBC022, is not MDR since it displays

resistance to less than three classes of antibiotics (Fig. 2b, Table 1). The other five strains tested showed resistance to three (AIEC in rich medium) or more classes of antibiotics in both media, including, as expected, the known MDR strain MH258 in addition to the clinical isolates from hospitalized patients Ec1898, Kp1012, and Kp834 (Fig. 2b, Table 1), supporting their clustering together as predicted MDR strains (Fig. 1d). Most of all, these results endorse that ARO112 is not MDR and highlight that bacterial phenotypes are context dependent, with minimal medium frequently potentiating resistance phenotype.

Biofilms are microbial communities enclosed in extracellular matrices and adherent to a substrate, and can contribute to increased recalcitrance to antibiotic treatments[46]. We found that Kp1012 was the strain that produced the most robust biofilm biomass in minimal medium. In rich medium, most strains resulted in low (EcN, Kp834, MBC022, ARO112) or very low (Ec1898, MH258) biofilm biomass (Fig. 2c, Kruskal-Wallis' test with Dunn's correction for multiple comparisons, p < 0.05), with only MG1655, AIEC, and Kp1012 showing slightly higher biofilm capacity (Fig. 2c, Kruskal-Wallis' test with Dunn's correction for multiple comparisons, p < 0.05). Under minimal medium, ARO112 presented some capacity for biofilm formation but much lower than Kp1012 (Fig. 2c, Kruskal-Wallis' test with Dunn's correction for multiple comparisons, p < 0.05).

Urease production is a common trait in *K. pneumoniae* and *K. oxytoca* and can be problematic in urinary tract infections, whilst it might also be a good competitive trait for gut colonization[47]. In cultures grown in rich medium, there was no detectable urease activity, except for *K. oxytoca* MBC022 and, to a lesser extent, ARO112, both displaying low levels of urease (Fig. 2d). In cultures grown in minimal medium, all *Klebsiella* strains, but not *E. coli* strains, showed significantly higher urease activity, with Kp1012 and MBC022 presenting the highest. ARO112, together with Kp834 and MH258, had the lowest activities, with ARO112 displaying only a small increase in comparison to rich medium (Fig. 2d, Kruskal-Wallis' test with Dunn's correction for multiple comparisons, p < 0.05). The absence of urease activity in the *E. coli* strains is expected since this species is a known non-urease producer, with only some clinical isolates of pathogenic Enterohemorrhagic *E. coli* reported to produce urease[48]. Strains proficient in both biofilm formation and urease production, as is the case for Kp1012, can become problematic and often associated with recurrent urinary tract infections[49]. Notably, ARO112, showed low biofilm and low urease production capacity.

While in medium not limited in iron (rich medium), all strains showed a reduced production of iron chelators, siderophores (Fig. 2e). Importantly, in the iron-limiting (M9+glucose) medium, siderophore production was clearly low in ARO112 (as well as in *E. coli* MG1655, and *K. oxytoca* MBC022). This occurred even though growth yield was unaffected for ARO112, when comparing rich and minimal media (Supplementary Fig. 1b). In contrast, all pathogenic strains produced high levels of siderophores in this medium (Fig. 2e, Kruskal-Wallis' test with Dunn's correction for multiple comparisons, *p* < 0.05).

When testing resistance to serum-mediated killing of bacteria grown in rich medium, only the *K. pneumoniae* strains and MBC022 showed significant resistance to and replication ability in the presence of human serum (Fig. 2f), indicating their potential to survive and thrive in the bloodstream of patients. On the contrary, most *E. coli* strains (MG1655, AIEC, EcN) are very susceptible to human serum (Kruskal-Wallis' test with Dunn's correction for multiple comparisons, p < 0.05), while Ec1898 and ARO112 are not killed nor do they replicate in human serum (Fig. 2f). Interestingly, testing these strains grown in minimal medium instead resulted in a significantly higher replication for MH258, increasing from 40- (in rich medium) to 140-fold (Kruskal-Wallis' test with Dunn's correction for multiple comparisons, p < 0.05). More surprising, however, was the change detected for EcN, where we observed a strong serum resistance and replication capacity when grown in minimal medium (70-fold increase; Fig. 2f, Kruskal-Wallis' test

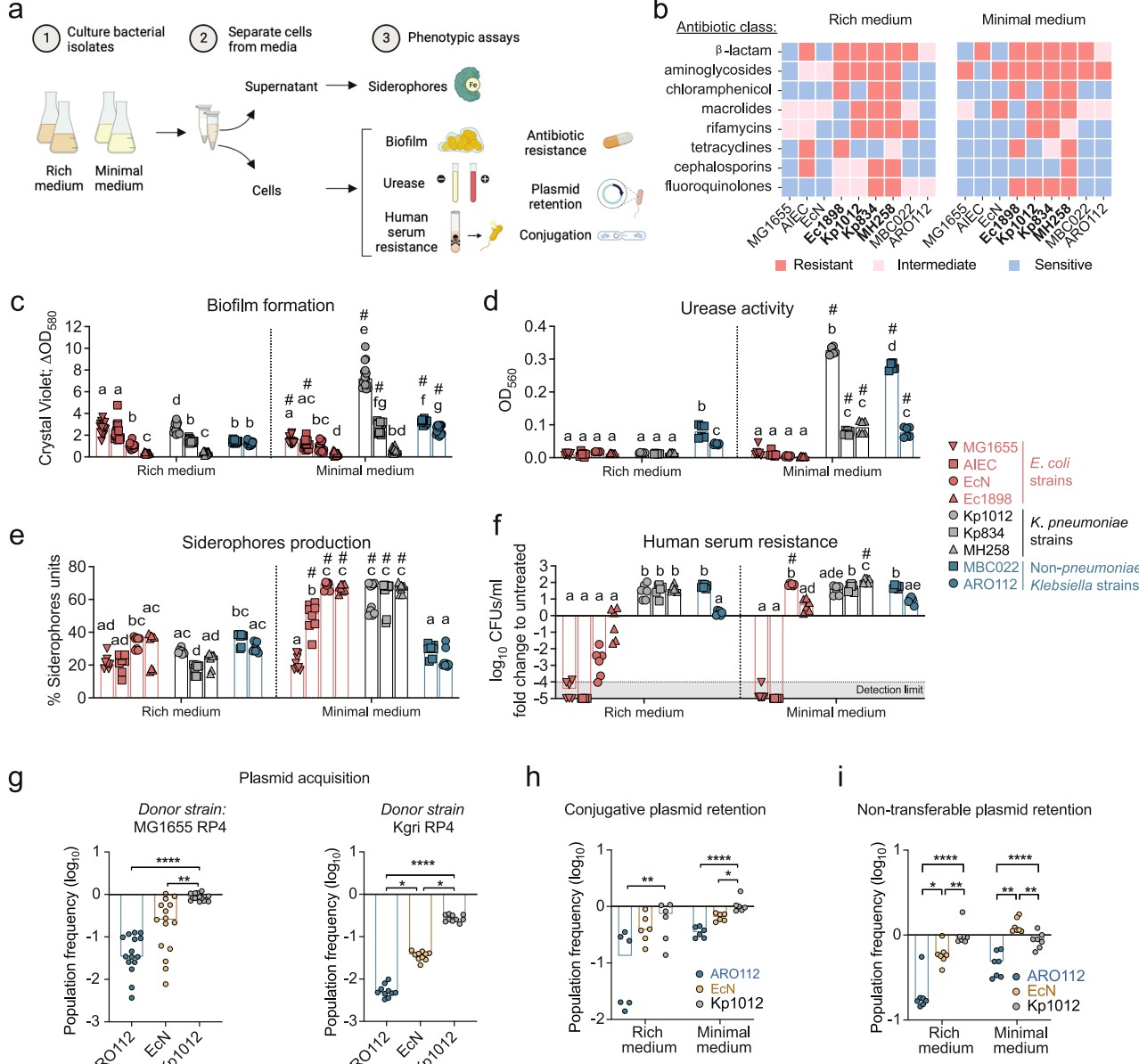

**Fig. 2 | Phenotypical evaluation of ARO112 and other Enterobacteriaceae strains for nosocomial-related pathogenic factors important for host invasion and colonization. a** Schematic representation of the experimental setup used to test bacterial strains for pathogenic traits. A set of 9 strains (4 *E. coli*, 3 *K. pneumoniae*, 2 non-*pneumoniae Klebsiella*) were tested, both in rich and minimal media, for: **b** antibiotic resistances, MDR Enterobacteriaceae strains are represented in bold (resistant to >3 classes of antibiotics); **c** biofilm formation; **d** urease activity; **e** siderophore production; and **f** capacity to resist killing or inhibition by human serum. Selected strains (ARO112, EcN, and Kp1012) were tested for their capacity to **g** acquire the conjugative plasmid RP4 from MG1655 donor strain or Kgri donor strain, or **h** retain the plasmid in rich and minimal media. **i**, the capacity to retain the non-self-transmissible plasmid pMP7605 in rich and minimal media was also tested. In (**c**–**i**} bars represent median values. **b** includes data from a total of 6 replicates per group, from two independent experiments; **c** from a total of 15 replicates per

group, from three independent experiments; **d** from a total of 6 replicates per group, from two independent experiments; **e** more than 8 replicates per group, from three independent experiments; **f** from a total of 6 replicates per group, from two independent experiments; **g** from a total of 15 (from MG1655 RP4) or 10 (From Kgri RP4) replicates per group, from 2-4 independent experiments; **h**, from a total of 6 replicates per group, from 2 independent experiments; **i** a total of 7 replicates per group, from 2 independent experiments. In **c**–**g** statistical analyses were done using Kruskal-Wallis' test with Dunn's correction for multiple comparisons. In **c**-**f** different letters denote significant differences among strains within the same medium (p < 0.05) and # denotes significant differences between different media for the same strain (p < 0.05). In **h** and **i** data were analyzed using Two-way ANOVA with Sidak's correction for multiple comparisons. * *p* < 0.05; ** *p* < 0.01; **** *p* < 0.0001. Panel a was adapted from a figure created in BioRender. Oliveira, R. (2025) https://BioRender.com/m6yvg7v.

with Dunn's correction for multiple comparisons, p < 0.05). Reports on this phenotype are usually performed using rich media to grow bacteria[50,51], but here, we found that some bacteria, like EcN and MH258, show increased proliferation when grown in minimal medium. Finally, ARO112 shows no evidence of being killed by the human serum, but neither does it replicate to the levels seen for most strains (Fig. 2f).

Since this pathogenic trait directly affects the host, as it potentially allows bacteria to proliferate in the bloodstream, with life-threatening risks, ARO112's inability to proliferate is a desirable trait for a potential probiotic.

Here, we established a comprehensive protocol for the evaluation of pathogenic traits that can be used to characterize the probiotic

**Table 1 | Antibiotics and respective concentrations used**

| Class | Antibiotic | Concentration (µg/ml) |
|---|---|---|
| β-lactams (penicillins) | ampicillin | 100 |
| | carbenicillin | 50 |
| Aminoglycosides | streptomycin | 100 |
| | kanamycin | 50 |
| | gentamicin | 15 |
| | neomycin | 200 |
| | apramycin | 50 |
| | spectinomycin | 50 |
| Chloramphenicol | chloramphenicol | 25 |
| Macrolides | erythromycin | 125 |
| Rifamycins | rifampicin | 20 |
| Tetracyclines | tetracycline | 10 |
| Quinolone | nalidixic acid | 32 |
| Cephalosporins | cefotoxitin | 8 |
| Fluoroquinolones | ciprofloxacin | 0.5 |

potential of non-*pneumoniae Klebsiella* strains. The characterization of the pathogenic potential based on phenotypic assays gives us the resolution to the strain level for the pathogenic traits selected. While the genome-based approach showed us that the pathogenic traits predicted to be present or absent in the genome correlate with the taxonomy, the selective testing of clinically-important phenotypes allows us to distinguish distinctly active pathogenic traits within a taxon, such as the heterogeneity within both the *E. coli* and the *K. pneumoniae* clades, but also to pinpoint strains with overall less pathogenic traits associated with nosocomial infections. Notably, ARO112 stands out with low pathogenic traits commonly associated with nosocomial infections meeting the baseline criteria for safeness established by EcN[52].

## ARO112 has low efficiency for acquiring and retaining antibiotic resistance by Horizontal Gene Transfer

The lack of resistance found in the commensal non-clinical strains does not exclude the potential for antibiotic resistance acquisition by HGT, which is common among Enterobacteriaceae within complex communities, like the gut microbiota[53,54]. Indeed, the acquisition of antibiotic resistance, in particular by HGT events, has been shown to be problematic, with Enterobacteriaceae at the forefront of this increasingly concerning medical issue[45]. Therefore, taking into account that ARO112 seems to lack for genes predicted to be related to conjugation (Fig. 1e), we assessed the capacity of ARO112 to acquire resistances from donor bacteria through HGT, in comparison with the probiotic EcN and the MDR *K. pneumoniae* strain Kp1012. We tested the transfer of a commonly used conjugative plasmid (RP4) from a donor strain (*E. coli* MG1655 or *K. grimontii*) into the aforementioned recipient strains. Interestingly, the already MDR strain Kp1012 had the highest acquisition rate, with a median of 83% (Fig. 2g, Kruskal-Wallis' test with Dunn's correction for multiple comparisons, p < 0.0001 (compared to ARO112) and p = 0.0067 (compared to EcN)) and 25% (Fig. 2g, Kruskal-Wallis' test with Dunn's correction for multiple comparisons, p < 0.0001 (compared to ARO112) and p = 0.0332 (compared to EcN)) of cells acquiring the plasmid from the *E. coli* and *K. grimontii* donors, respectively. EcN presented an intermediate transfer rate, with a median of 25 and 4% (Fig. 2g, Kruskal-Wallis' test with Dunn's correction for multiple comparisons, p = 0.0332 (compared to ARO112)) acquiring the plasmid and consequent antibiotic resistance from *E. coli* and *K. grimontii*, respectively. In contrast, ARO112 showed strong resistance to plasmid acquisition, with a median of less than 4% of the population acquiring the RP4 plasmid from *E. coli*, and less than 0.5% from *K. grimontii* (Fig. 2g). Next, we tested the capacity to retain the

acquired plasmids, after five days of daily passages in rich or minimal media in the absence of antibiotic selection. Remarkably, ARO112 showed the highest plasmid loss, with 87% (Fig. 2h, two-way ANOVA with Sidak's correction for multiple comparisons, p = 0.0059 (compared to Kp1012)) and 65% (Fig. 2h, two-way ANOVA with Sidak's correction for multiple comparisons, p < 0.0001 (compared to Kp1012)) of the population losing the plasmid in rich and minimal media, respectively. EcN followed, with 61% and 34% (Fig. 2h, two-way ANOVA with Sidak's correction for multiple comparisons, p = 0.0186 (compared to Kp1012)) of the population losing the plasmid under the same condition. In contrast, the Kp1012 lost the plasmid in only 27% and 2% of the population in rich and minimal media, respectively (Fig. 2h).

Even plasmids that are non-self-transmissible, can be transferred among bacteria, through conjugation events[55] in which non-conjugative plasmids take advantage of conjugative plasmids' machinery. Therefore, we also tested the capacity of ARO112, EcN, and Kp1012 to retain a non-self-transmissible plasmid (pMP7605) carrying an antibiotic resistance received by electroporation. Kp1012 barely lost the plasmid, both in rich (7%, two-way ANOVA with Sidak's correction for multiple comparisons, p < 0.0001 (compared to ARO112) and p = 0.0039 (compared to Ecn)) and minimal media (11%, two-way ANOVA with Sidak's correction for multiple comparisons, *p* < 0.0001 (compared to ARO112) and p = 0.0047 (compared to Ecn); Fig. 2i), and 42% (two-way ANOVA with Sidak's correction for multiple comparisons, *p* = 0.0103 (compared to ARO112)) of EcN colonies lost the non-self-transmissible plasmid in rich medium (Fig. 2i), while no loss was observed in minimal medium (Fig. 2i, two-way ANOVA with Sidak's correction for multiple comparisons, *p* = 0.006 (compared to ARO112)). Remarkably, ARO112 showed the highest plasmid loss, with 83 and 51% of the population losing the non-self-transmissible plasmid when grown in rich or minimal media, respectively (Fig. 2i).

Natural plasmids are known to be more stably maintained as they often confer some fitness advantages that drive their persistence[56]. We also tested the retention capacity of Kanamycin resistance transferred from one of the MDR clinical isolates predicted to bear natural plasmid(s), Kp834, into ARO112 and EcN (Kp1012 was not tested due to its MDR status, complicating counterselection). EcN did not lose the resistance, while ARO112 lost the resistance in over 15% (rich medium) and 30% (minimal medium) of the population (Supplementary Fig. 1c).

In summary, even though ARO112 shares with other Enterobacteriaceae strains the capacity to acquire plasmids, when compared to the probiotic strain EcN and the *K. pneumoniae* clinical isolate Kp1012, ARO112 is much less efficient in plasmid acquisition and retention, a main source of antimicrobial resistance in Enterobacteriaceae, thus reaffirming the safety benchmarks established by EcN and, therefore, presents a reduced risk of becoming a clinical problem.

## ARO112 safeness is maintained upon gut colonization

Having observed that the tested phenotypes varied depending on the culture medium used, we questioned which of the tested laboratory media, rich or minimal, was resembling the phenotypical profiles of bacteria in the mammalian gut more closely. Therefore, we tested the phenotypes whose assays do not require bacterial growth, and which could be tested directly in bacteria collected from fecal samples of colonized mice (Fig. 3a). We compared the phenotypes of ARO112 with EcN and Kp1012 collected from fecal samples of ex-germ-free mice colonized for five days with each of these bacteria, without selection or growth. Regarding urease, Kp1012 presented the same high urease activity detected in minimal medium-grown cultures (Kruskal-Wallis' test with Dunn's correction for multiple comparisons, *p* = 0.0014 (compared to EcN) and EcN showed the expected low urease activity also observed in both laboratory-grown cultures (Kruskal-Wallis' test with Dunn's correction for multiple comparisons, p = 0.0451 (compared to ARO112), Figs. 3b, 2d); ARO112 presents a variable phenotype,

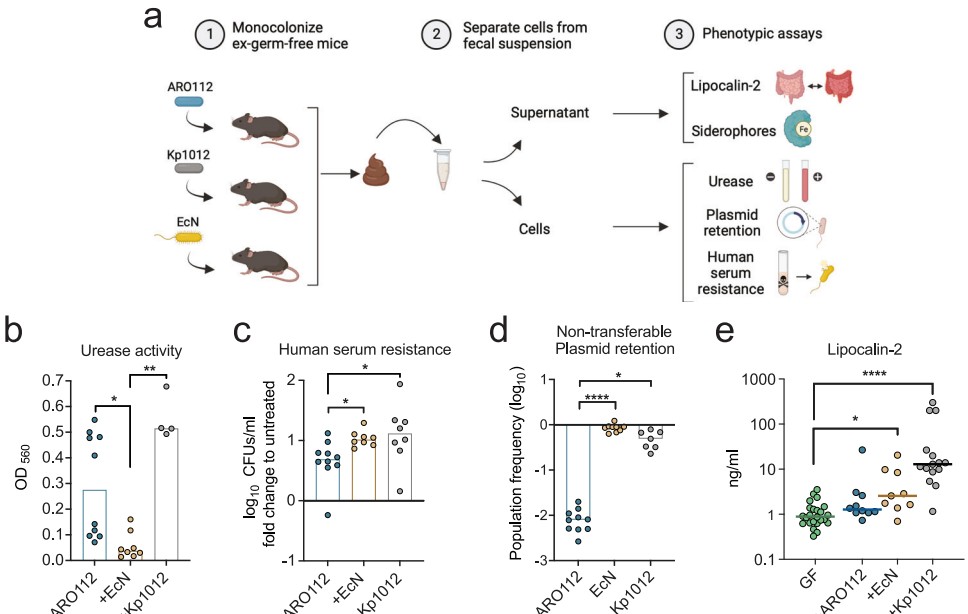

**Fig. 3 | Phenotypic testing of ARO112, EcN, and Kp1012 isolated from fecal samples of mono-colonized mice. a** Schematic representation of the experimental setup used to test bacterial strains for pathogenic traits after gut colonization. Germ-free (GF) mice were mono-colonized with each of selected strains (ARO112, EcN, Kp1012) for five days, after which bacteria were isolated from the fecal samples and tested for **b** urease activity (ARO112 n = 10, EcN n = 8, Kp1012 n = 4, two independent experiments) and **c** resistance to human serum (ARO112 n = 10, EcN n = 8, Kp1012 n = 8, two independent experiments). **d** Bacteria were also evaluated for plasmid retention upon intestinal colonization (ARO112 n = 10, EcN

n = 9, Kp1012 n = 7, three independent experiments). **e** Fecal supernatants were evaluated for Lipocalin-2/NGAL levels, as a marker for intestinal inflammation. Lcn2 was tested in germ-free animals before colonization (GF, N = 24), and after colonization with ARO112 (N = 10), EcN (N = 9), or Kp1012 (N = 15), from at least three independent experiments. In (**b**–**e**), data were analyzed using Kruskal-Wallis' test with Dunn's correction for multiple comparisons (* $p < 0.05$; ** $p < 0.01$; **** $p < 0.0001$). In (**b**–**e**), bars and lines represent median values. Panel a was adapted from a figure created in BioRender. Oliveira, R. (2025) https://BioRender.com/r57ge77.

with half the samples showing low urease activity, similar to the results obtained with cultures grown in the laboratory, while the other half showed increased urease activity (Figs. 3b, 2d). This result, while surprising due to the low levels of urease activity in culture-grown ARO112, indicates that in this strain this phenotype might be regulated differently in the gut environment.

Regarding fecal siderophore measurements, the values obtained were low for all three strains, but with a similar tendency observed in minimal medium of lower values for ARO112, when compared with both EcN and Kp1012 (Supplementary Fig. 2a and Fig. 2e).

When testing for resistance to human serum after gut colonization, similar results were obtained with bacteria extracted directly from feces of mono-colonized mice (Fig. 3c) to the ones of laboratory-grown cultures, namely minimal medium (Fig. 2f). Importantly, ARO112 strain, even after colonizing the mouse gut, was more sensitive to human serum than both Kp1012 (Kruskal-Wallis' test with Dunn's correction for multiple comparisons, p = 0.0270) and EcN (Fig. 3c, Kruskal-Wallis' test with Dunn's correction for multiple comparisons, p = 0.0485).

We also checked the non-self-transmissible plasmid retention in the murine gut by testing ex-germ-free mice colonized with each of these strains carrying the non-self-transmissible plasmid pMP7605, and quantifying how much of the population retained the plasmid after five days of colonization. EcN lost the plasmid in only 11% (Kruskal-Wallis' test with Dunn's correction for multiple comparisons, p < 0.0001 (compared to ARO112)) of the population, while Kp1012 lost in 51% (Kruskal-Wallis' test with Dunn's correction for multiple comparisons, p = 0.0391 (compared to ARO112), Fig. 3d). Remarkably, ARO112 lost the plasmid in more than 99% of cells (Fig. 3d), showing that the recalcitrance of ARO112 to maintaining acquired antibiotic resistances is potentiated in the murine gut when compared with laboratory-grown cultures.

The ex-germ-free mouse model also allowed us to test the host reaction to the colonization by each of the three aforementioned strains, by assessing the fecal levels of Lipocalin-2 (Lcn2), as a common fecal marker for gut inflammation[57,58]. After 5 days of mono-colonization, fecal levels of Lcn2 in mice colonized with ARO112 were similar to germ-free mice, providing no evidence for ARO112 inducing intestinal inflammation in mono-colonized mice (Fig. 3e Kruskal-Wallis' test with Dunn's correction for multiple comparisons). Mice colonized with EcN presented a slight increase in Lcn2 levels (Kruskal-Wallis' test with Dunn's correction for multiple comparisons, p = 0.0249). In contrast, colonization with Kp1012 induced considerably higher levels Lcn2 when compared to the germ-free mice levels (Fig. 3e, Kruskal-Wallis' test with Dunn's correction for multiple comparisons, p < 0.0001). Next, to further characterize other host-related response parameters to ARO112 colonization, namely body temperature and weight, blood glucose levels, and the capacity or lack-thereof to translocate and colonize extra-intestinal organs, we repeated the mono-colonization experiment with ARO112 and compared these host-responses with those observed with mice mono-colonized with *S.* Typhimurium, which is able to translocate and cause disease. Mice mono-colonized with *S.* Typhimurium started showing signs of disease 12-18 h post-gavage with decreased body weight, temperature and glucose levels (Supplementary Fig. 2b–d). Therefore, after 18 h of mono-colonization with *S.* Typhimurium, mice were sacrificed and dissected to assess organ translocation. *S.* Typhimurium organ translocation was observed in all mice, with CFUs detected in at least one extra-intestinal organ (spleen, kidneys, liver, or lungs) of each of these animals (Supplementary Fig. 2e). In contrast, animals mono-colonized with ARO112 showed no signs of disease even 5 days after colonization, with no changes in body weight, temperature, or glucose levels (Supplementary Fig. 2b–d). Additionally, despite the intestinal colonization loads of *S.*

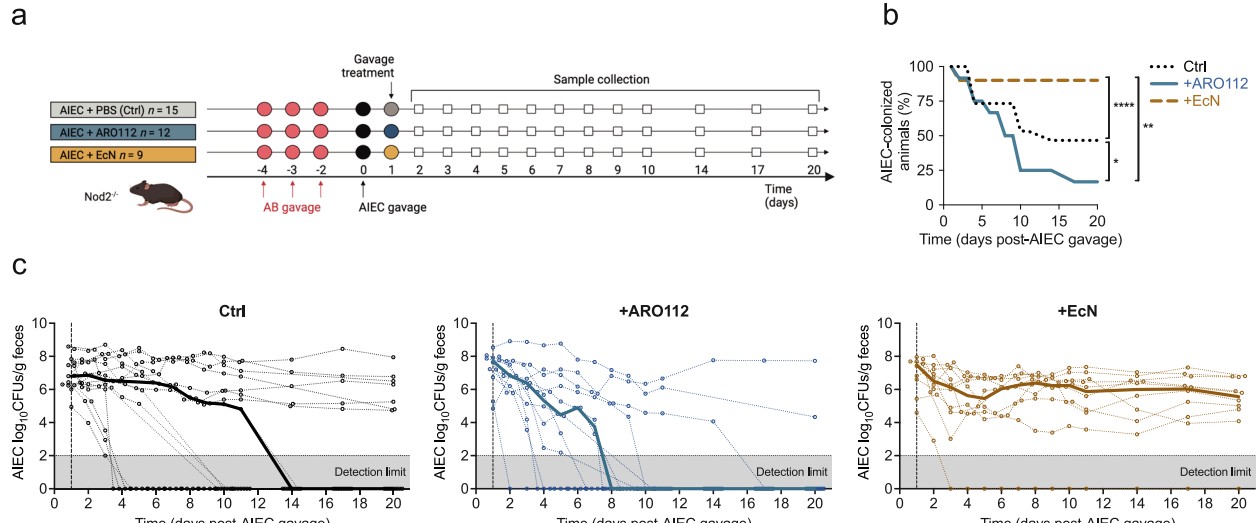

**Fig. 4 | Evaluation of therapeutical potential capacities of ARO112 and EcN against AIEC colonization in an IBD mouse model with antibiotic treatment.** **a** Experimental setup of IBD mouse model Nod2$^{-/-}$ mice treated with antibiotics, followed by AIEC infection and treated with ARO112, EcN or PBS (Ctrl). **b** Percentage of mice colonized with AIEC in the different groups throughout the experiment. **c** AIEC loads during the experiment. Thin lines represent AIEC loads in individual mice over time, while thick lines represent median values for each treatment over time. In (**b** and **c**), data were analyzed using Two-way ANOVA with Tukey's correction for multiple comparisons (* $p < 0.05$; ** $p < 0.01$; **** $p < 0.0001$). AIEC fecal loads and colonization were tested in a total of 15 (Ctrl), 12 ( + ARO112), or 10 ( + EcN) animals, from 2 to 3 independent experiments. Panel a was adapted from a figure created in BioRender. Oliveira, R. (2025) https://BioRender.com/k6l9wp0.

Typhimurium and ARO112 being comparable, no bacterial extra-intestinal organ translocation was observed in the animals colonized with ARO112 (Supplementary Fig. 2e).

Together, these results indicate that ARO112 could be a safe next-generation probiotic since it has reduced resistance to human serum and low pathogenic potential. Moreover, it has low plasmid acquisition and retention capacities, as opposed to other Enterobacteriaceae strains. Additionally, ARO112 showed no evidence of inducing host inflammation, nor host-related disease signs.

## ARO112 promotes a faster recovery from AIEC gut infection in an IBD mouse model

Having shown that ARO112 has less potentially harmful features than an MDR *K. pneumoniae* and is comparable to the probiotic EcN, along with previous findings indicating that ARO112 is able to partially displace *E. coli* MG1655 from the gut of mono-colonized and antibiotic-treated mice[37], we tested the potential protective role of ARO112 against AIEC in a mouse model for IBD. AIEC is a pathobiont that can be problematic in IBD patients[59]. Thus, we studied the outcome of AIEC infections in Nod2$^{-/-}$ mice. Mutations in Nod2 are the most common mutations in patients genetically prone to IBD, and Nod2$^{-/-}$ mice, like IBD patients, have increased susceptibility to infections and lower capacity to recover from intestinal inflammation[60,61]. To test the outcome of AIEC infections after antibiotic treatment, we needed an antibiotic regimen that would allow AIEC colonization. Previous work by Drouet and colleagues[62] has shown that oral administration of two non-absorbed antibiotics (also used in clinical trials on IBD patients[63]), vancomycin and gentamicin, to Nod2$^{-/-}$ mice resulted in a longer-lasting AIEC colonization compared to WT animals, without lasting inflammatory effects in this antibiotic-treated infection model. We started by comparing susceptibility of WT and Nod2$^{-/-}$ mice to colonization by AIEC with this antibiotic regimen in our animal facility. All mice were treated by oral gavage with a combination of vancomycin and gentamicin, once daily for three consecutive days[62] (Fig. 4 and Supplementary Fig. 3a). After a washout period treatment of 2 days following antibiotic, mice were gavaged with AIEC, and AIEC colonization was followed through selective plating. This approach differed from that of Drouet and

colleagues[62], who infected mice on the last day of antibiotic treatment, as we wanted to avoid antibiotics interfering with AIEC infection and the host. Consistent with previous observations[62], Nod2$^{-/-}$ mice were more susceptible to AIEC colonization in comparison with WT mice, with ~50% of the Nod2$^{-/-}$ mice still being colonized with AIEC after 20 days of infection (Fig. 4b, control animals), while all WT mice had cleared the infection in that same timeframe (Supplementary Fig. 3a, control mice).

Next, to test the therapeutic potential of ARO112, one day following AIEC infection, Nod2$^{-/-}$ mice were gavaged with either ARO112 strain or the probiotic EcN strain (Fig. 4a). AIEC clearance was faster in mice treated with ARO112, with 50% of the animals being AIEC-free after 8 days of infection (7 days of probiotic treatment), compared to mice that received no probiotic treatment ( + PBS, control mice, two-way ANOVA with Tukey's correction for multiple comparisons, p = 0.0313), in which 14 days were needed for 50% of the mice to resolve the infection, with 7 of 15 mice still infected at the end of treatment (Fig. 4b, c). We also tested the effect of the probiotic strain, EcN, as it has been successfully used against AIEC infections in zebra-fish and human intestinal epithelial cell lines[64–66] and also in an ulcerative colitis clinical trial[30,31]. In this model, treatment with the probiotic EcN did not help AIEC decolonization, with all but one tested mice remaining colonized with high levels of AIEC for the 20 days of the infection (Fig. 4b, c, two-way ANOVA with Tukey's correction for multiple comparisons, p < 0.0001 (compared to Ctrl) and p = 0.0035 (compared to +ARO112)). We highlight that EcN failed to colonize the one mouse of this group that displaced AIEC infection and therefore, the displacement observed in this animal cannot be attributed to the EcN treatment (Supplementary Fig. 3b). When assessing the number of mice that resolve the infection in each condition, it is noticeable that treatment with EcN unexpectedly promoted a continued infection with AIEC, while the absence of probiotic treatment led to the infection being resolved in half of the population (Fig. 4b, c), thus the outcome obtained with the treatment with ARO112, in which 10 of 12 mice (>83%) became AIEC-free by the end of the experiment, was the most efficient treatment tested (Fig. 4b). Additionally, we observed that ARO112 is itself cleared throughout the experiment, after the clearance of AIEC, having ultimately disappeared in 9 out of the 12 mice tested

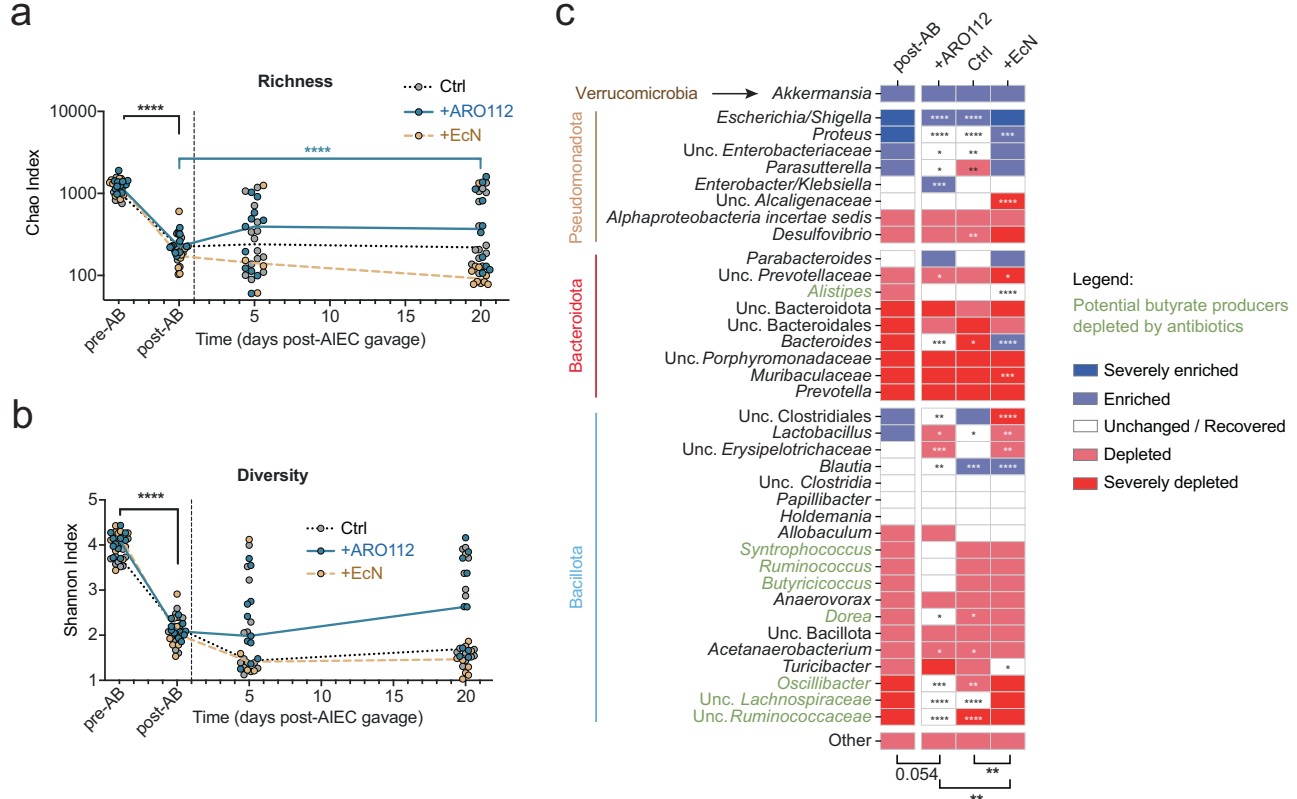

**Fig. 5 | Intestinal microbiota composition resultant from treatment with probiotic strains (ARO112, EcN) or lack thereof (Ctrl). a** Richness and **b** diversity of gut microbiota in fecal samples collected before (pre-AB, day -4), or after (post-AB, day 0) antibiotic treatment and 5 and 20 days after AIEC infections of mice treated with ARO112, EcN or PBS (Ctrl). In (**a** and **b**), lines represent median values for each treatment over time. **c**, Changes in the relative abundances of most prevalent taxa at day 20 after AIEC infection in comparison with the levels before antibiotics (day -4). In (**a**−**c**), data were analyzed using Two-way ANOVA with Dunnett's correction for multiple comparisons (*$p < 0.1$; **$p < 0.01$; ***$p < 0.001$; ****$p < 0.0001$). Fecal microbiota composition analyses were performed in a total of 14 (Ctrl), 10 (+ ARO112), and 4 (day 5) or 10 (days 0,-4, 0, 20) (+ EcN) samples, from 2-3 independent experiments.

(75%) by the end of the experiment (Supplementary Fig. 3b). Importantly, the protective effect of ARO112 observed in Nod2$^{-/-}$ mice was not replicated in WT mice treated with the same antibiotic regimen; in these animals spontaneous AIEC decolonization occurred faster than in the Nod2$^{-/-}$ and at the same rate with and without probiotic treatment (Supplementary Fig. 3a).

We have previously shown that ARO112 could partially displace MG1655 in streptomycin-treated WT mice[37] by a direct interaction between these two species through nutrient competition that could be observed both in ex-germ-free animals and in laboratory cultures[37]. Consistent with these results, in the absence of microbiota in monocolonized WT or Nod2$^{-/-}$ mice with AIEC, a partial displacement promoted by ARO112 was observed (~4 fold in WT (p = 0.0161) and ~16 fold in Nod2$^{-/-}$ (p < 0.0001); Supplementary Fig. 3c−e, Two-way ANOVA with Sidak's correction for multiple comparisons). This displacement was also comparable to what was observed when co-colonized with other Enterobacteriaceae, like the MDR clinical isolates Ec1898 (>4 fold, *p* = 0.0317) and Kp1012 (~6 fold, p = 0.019); Supplementary Fig. 3f, g, Mann-Whitney test), and previously with MG1655[37] ( >10 fold) or against other Pseudomonadota species like *Vibrio cholerae* (>38 fold, p = 0.0007; Supplementary Fig. 3h, Mann-Whitney test). These results showing that ARO112 only partially inhibits AIEC in the absence of other microbiota members (mono-colonized germ-free mice), indicate that direct competition between ARO112 and AIEC is not the main mechanism driving clearance in the SPF antibiotic treated Nod2$^{-/-}$ animals.

Altogether, these results show that ARO112 can accelerate full clearance of AIEC in the Nod2$^{-/-}$ animals only in the presence of other members of the microbiota and the partial displacement capacity of ARO112 towards AIEC-and other Pseudomonadota pathogens in the absence of a microbiome highlights a possible synergistic effect of ARO112 with the gut microbiota.

### ARO112 treatment promotes microbiota recovery, namely of potential butyrate-producing bacteria depleted by antibiotic treatment

To further understand the effect of ARO112 on AIEC in the IBD model, we analyzed the fecal microbiota composition of Nod2$^{-/-}$ mice by 16 s rRNA gene sequencing before (pre-AB) and after (post-AB) antibiotic treatment, as well as post-AIEC colonization (day 5) and after 20 days of experiment upon treatments with PBS (Ctrl), EcN, or ARO112. We observed that mice treated with ARO112 showed a significant recovery in richness (*p* = 0.0335) and a similar tendency in diversity, unlike the long-lasting decrease of both metrics in the mice from the other groups, which persisted at lower levels after the antibiotic treatment (Fig. 5a, b, two-way ANOVA with Dunnet's correction for multiple comparisons).

We then analyzed how antibiotic treatment affected the most abundant taxa (37 taxa representing an average of 98% abundance of all taxa; Fig. 5c). Beta-diversity analysis using Bray-Curtis dissimilarity index, performed through Principal Coordinates Analysis (PCoA) on the 37 taxa, shows that antibiotic treatment significantly alters the microbiota (pre-AB vs post-AB, Supplementary Fig. 4a, one-way PERMANOVA with 999 permutations with Bonferroni corrections, p < 0.05). Out of these 37 taxa, 22 (59%) were depleted by the antibiotics, while 7 (19%) were enriched and 8 (22%) remained unchanged

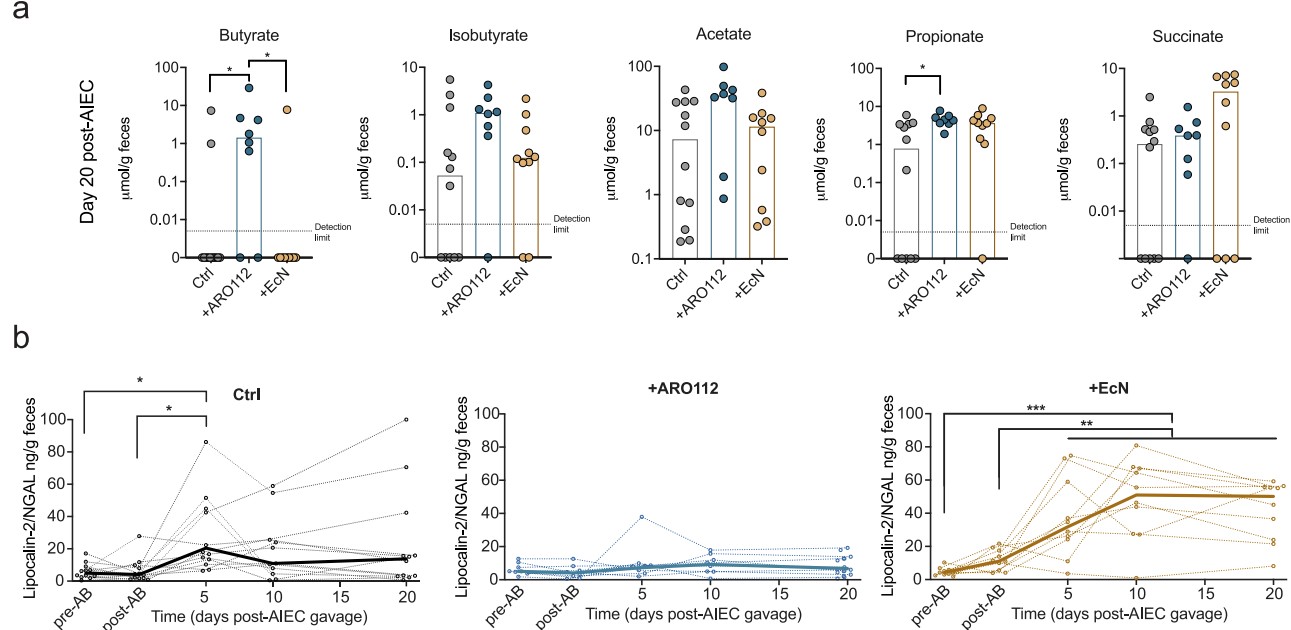

**Fig. 6 | SCFA levels and intestinal inflammation profiles of mice upon different probiotic treatments (ARO112, EcN) or lack thereof (Ctrl). a** Metabolomic profile of butyrate, isobutyrate, acetate, propionate, and succinate in fecal samples of IBD mouse model Nod2$^{-/-}$ mice treated with ARO112, EcN or PBS (Ctrl), at the end of the experiment (day 20). **b** Lipocalin levels (Lcn2) in fecal supernatants were measured as a marker for intestinal inflammation throughout the experiment. Thin lines represent Lcn2 levels in individual mice overtime, while thick lines represent median values for each treatment overtime. In (**a**), bars represent median values and data were analyzed using Kruskal-Wallis test with Dunn's correction for multiple comparisons. In (**b**), data were analyzed using Two-way ANOVA with Tukey's correction for multiple comparisons (* $p < 0.05$; ** $p < 0.01$; *** $p < 0.001$; **** $p < 0.0001$). Metabolites and Lcn2 were measured in 12 (Ctrl), 8 ( + ARO112) and 10 ( + EcN) samples.

by the end of the experiment. At the end of the experiment (day 20), the microbiota composition of mice treated with either PBS, EcN, or ARO112 remained different from pre-AB (Supplementary Fig. 4a, one-way PERMANOVA with 999 permutations with Bonferroni corrections, $p < 0.05$), however, treatment with ARO112 resulted in samples compositions positioning significantly closer to their pre-AB levels (Supplementary Fig. 4b, two-way ANOVA with Sidak's correction for multiple comparisons, $p = 0.0251$). More specifically, mice from the control group or treated with EcN were able to recover 14% (3 of 22) of taxa depleted by the antibiotic treatment (Fig. 5c, two-way ANOVA with Dunnett's correction for multiple comparisons, $p < 0.05$). Consistently with the microbiota beta and alpha diversity recovery data, more taxa recovered in the group treated with ARO112 strain, where mice were able to recover 41% (9 of 22) of taxa depleted by the antibiotic treatment (Fig. 5c, two-way ANOVA with Dunnett's correction for multiple comparisons, $p < 0.05$).

Analysis of the groups of bacteria affected by antibiotic treatment showed that 8 taxa of potential butyrate producers[67] were depleted by the antibiotics (highlighted in green in Fig. 5c, two-way ANOVA with Dunnett's correction for multiple comparisons, p < 0.05), from which 4 recovered upon 19 days of treatment with ARO112 (*Dorea*, Lachnospiraceae, *Oscillibacter*, and Ruminococcaceae), while only 1 was recovered in the control (Lachnospiraceae) and another with EcN treatment (*Alistipes*) (Fig. 5c, two-way ANOVA with Dunnett's correction for multiple comparisons, p < 0.05). Bray-Curtis dissimilarity indices for the 8 taxa of potential butyrate producers depleted by antibiotics also show that, even though both the PBS (p = 0.0008) and ARO112 (p < 0.0001) treatments resulted in microbiomes with a composition significantly closer to the pre-AB levels (Supplementary Fig. 4c, two-way ANOVA with Sidak's correction for multiple comparisons), only the population of samples treated with ARO112 is significantly similar to pre-AB samples in the PCoA plot, highlighting an overall consistent recovery of this group of taxa promoted by ARO112 (Supplementary Fig. 4d).

In summary, here we show that in a mouse model susceptible to infection, recovery of the dysbiotic microbiota, namely of microbiota members potentially involved in butyrate production, was improved in the animals treated with ARO112.

## Probiotic therapy with ARO112 enhances microbiota production of butyrate

As described in the above section, treatment with ARO112 leads to a better recovery of microbiota richness and diversity, including potential butyrate-producing bacteria. Several studies have shown that short-chain fatty acids (SCFAs), namely butyrate, is an important metabolite in IBD, with depletion of butyrate-producing gut bacteria by antibiotic treatments being considered a clinically-relevant concern in these patients[16,68]. For that reason, increasing butyrate production to improve IBD outcomes in patients has been previously proposed[69]. Therefore, our observation that ARO112 treatment promotes potential butyrate producers in the microbiota, possibly leading to an increased and sustained production of butyrate within the microbiota and gut epithelium, instead of the need for continuous exogenous administration, could be clinically relevant. Therefore, we measured SCFA concentrations including butyrate, isobutyrate (an isomer of butyrate), acetate, propionate, succinate in fecal samples of mice before antibiotic treatment (Supplementary Fig. 4e) and 19 days after treatment with probiotics or PBS (control; Fig. 6a). In agreement with our previous results showing the increase of potential butyrate-producing bacteria, treatment with ARO112 resulted in a clear increase of fecal levels of butyrate when compared to mice treated with PBS (*p* = 0.0253) or with EcN (*p* = 0.0197; Fig. 6a, Kruskal-Wallis with Dunn's correction for multiple comparisons), with most mice from the latter groups being depleted of butyrate at the end of the experiment (*p* = 0.0016 for Ctrl and p = 0.0003 for +EcN), while mice treated with ARO112 had their butyrate levels restored to pre-antibiotic treatment levels (Supplementary Fig. 4f, two-way ANOVA with Dunnett's correction for multiple comparisons). Isobutyrate, acetate, and succinate

levels did not significantly change across treatments, while propionate levels were significantly higher with ARO112 than with PBS ($p = 0.0163$), and similar to the levels in EcN group.

Since butyrate has been linked to intestinal health, with anti-inflammatory properties[70], we asked whether changes in this metabolite had an effect on intestinal inflammation by measuring lcn2 levels in fecal samples of the Nod2$^{-/-}$ mice infected with AIEC and treated with PBS, ARO112, or EcN. Lcn2 production is a host response to inflammation and infection and it is often used as a fecal marker for intestinal inflammation. Although Drouet and colleagues[62] did not observe signs of inflammation in their model, the phase-out AIEC infection from antibiotic treatment in our model could result in a different host response. Additionally, gut microbiota from mice with the same genetic background can greatly vary across facilities[71,72] and microbiota composition has been shown to influence host responses[73,74]. Notably, control animals exhibited a mild peak of Lcn2 levels during infection ($p = 0.0156$ compared to pre-AB and $p = 0.0189$ compared to post-AB), and treatment with EcN led to a persistent, albeit mild, increase in Lcn2 levels in this mouse model ($p < 0.001$ across timepoints compared to pre-AB and $p < 0.01$ compared to post-AB). In contrast, mice treated with ARO112 showed persistently low levels of Lcn2, never significantly varying from the levels prior to antibiotic treatment (Fig. 6b, two-way ANOVA with Tukey's correction for multiple comparisons).

These results show that ARO112 treatment not only led to increased pathogen clearance, but also to an increase in intestinal levels of butyrate and consistent low levels of intestinal inflammation throughout the course of AIEC infection in our mouse model. Although Lcn2 levels in this model were never high, the consistent Lcn2 levels below detection observed with ARO112 treatment—compared to the consistently mild levels observed with EcN treatment and transiently mild levels without treatment—suggest that ARO112 might have a protective effect against inflammation

## Non-pneumoniae Klebsiella strains are enriched in stool samples from IBD and non-IBD patients with low inflammation and high levels of potential butyrate producers

Having shown how ARO112 therapy promotes clearing of AIEC infection, recovery of intestinal butyrate production, and prevents mild inflammatory episodes upon infection in an antibiotic-treated murine model of IBD, we searched for evidence for the relevance of these processes in humans. For that, we took advantage of the publicly available data and metadata from the Human Microbiome Project (HMP2; https://www.ibdmdb.org/)[75]. This dataset includes IBD patients (diagnosed with Crohn's Disease – CD – or Ulcerative Colitis – UC) and participants not diagnosed with IBD (non-IBD; Supplementary Fig. 5a, b). To better compare our results on the effect of ARO112 post-antibiotic treatment with this database, we first excluded all samples collected under antibiotic treatment (Supplementary Fig. 5a, b). Additionally, because inflammation was measured in only a subset of samples/timepoints, we limited our analysis to samples that had both taxonomy and inflammation data (Supplementary Fig. 5a, b). These exclusions left us with a total of 460 samples (29%) from 93 participants (72%).

As expected, non-IBD participants display significantly lower levels of inflammation than IBD patients (CD and UC; Supplementary Fig. 5c, Kruskal-Wallis with Dunn's correction for multiple comparisons, $p < 0.0001$). Across all samples from all groups, we could discern three different levels of inflammation: high (>120 ng/ml), mild (>20 ng/ml), and low (<20 ng/ml) levels of fecal calprotectin, a marker for inflammation (Supplementary Fig. 5d, Kruskal-Wallis with Dunn's correction for multiple comparisons, p < 0.0001), a much higher range than what we observed in our IBD model with Lcn2. Since in our murine model, the inflammatory episode was stimulated by the E. coli AIEC, we mined the taxonomic data of this dataset to determine if the E. coli

abundance in the patients showed a relationship with their inflammation levels. In samples with mild or high levels of inflammation the loads of E. coli are significantly more abundant than in samples with low levels of inflammation (Supplementary Fig. 5e, Kruskal-Wallis with Dunn's correction for multiple comparisons), indicating a positive relationship between E. coli and inflammation.

Given the observed effects of ARO112 in our IBD mouse model, we examined the HMP2 dataset for potential relationships between non-pneumoniae Klebsiella species, similar to ARO112, and both butyrate-producing bacteria and inflammation. We identified 17 samples from 12 patients with detectable levels of non-pneumoniae Klebsiella, categorized as Klebsiella carriers, in contrast to 81 participants without detectable levels of non-pneumoniae Klebsiella (non-carriers). The low number of samples with detectable non-pneumoniae Klebsiella species is a limitation of this analysis, but it is consistent with the results from our previous study in mice where we had shown that non-pneumoniae Klebsiella species typically exist at very low levels in non-dysbiotic microbiomes and often fall below the detection threshold in taxonomic studies[37]. Despite this limitation, we found that in both non-IBD participants and patients with CD, carriers exhibited significantly higher levels of these potential butyrate producers (belonging to the families observed to be enriched in the ARO112-treated group in the murine experiment shown in Fig. 5c: Lachnospiraceae, Oscillospiraceae, Ruminococcaceae) than non-carriers (Supplementary Fig. 5f). Finally, within the 12 non-pneumoniae Klebsiella carriers (participants that had at least one sample positive for these bacterial species), we compared the inflammation levels between samples from these participants with and without detectable levels of non-pneumoniae Klebsiella. Notably, whenever non-pneumoniae Klebsiella species were detectable, inflammation was significantly lower (Supplementary Fig. 5g, Mann-Whitney test, p < 0.05). Both in human samples and in our antibiotic-treated murine model, non-pneumoniae Klebsiella species (i) positively correlate with potential butyrate producers and (ii) with lower levels of intestinal inflammation. This provides evidence that the protective role of non-pneumoniae Klebsiella species might also be relevant in humans.

## ARO112 treatment protects from disease development and intestinal inflammation in an IBD murine model of induced colitis

Given that IBD participants exhibit a much wider range of inflammation than what we observed in our IBD mouse model, we decided to assess ARO112's therapeutic effect under conditions that induce high inflammation, independently of infection. We used the established chemically-induced colitis model by providing DSS to the IBD mouse model Nod2$^{-/-}$. To induce chronic inflammation and colitis, we provided 2% DSS supplemented in the drinking water for 4 days (days-9 to −5), followed by a 4-day washout period with non-supplemented water (days −5 to 0), and a second cycle of 7 days with 2% DSS supplemented in the drinking water (days 0 to 7) with a second washout period with non-supplemented water for 7 days (days 7 to 15). Paired-littermates were divided into two groups, one treated with $10^8$ CFUs of ARO112 and the control group with the same volume of sterile PBS, at days −4, 4, 8, and 9 of the experiment (Fig. 7a).

We've assessed the disease activity index (DAI) considering: 1) weight loss, 2) stool consistency, and 3) rectal bleeding (Supplementary Data 6). A maximum subscore of 4 in each of these categories (i.e. a total DAI of 12), is treated as an endpoint for this experiment, due to loss of over 20% body weight and severe diarrhea with hematochezia (gross rectal bleeding), at which point mice were euthanized to avoid unnecessary suffering. Treatment with ARO112 protected all animals from reaching the maximum DAI, unlike animals in the control group (Fig. 7b, two-sided mixed-effect analysis with Sidak's correction for multiple comparisons, $p < 0.0001$). Even though all animals, regardless of treatment, lost body weight during

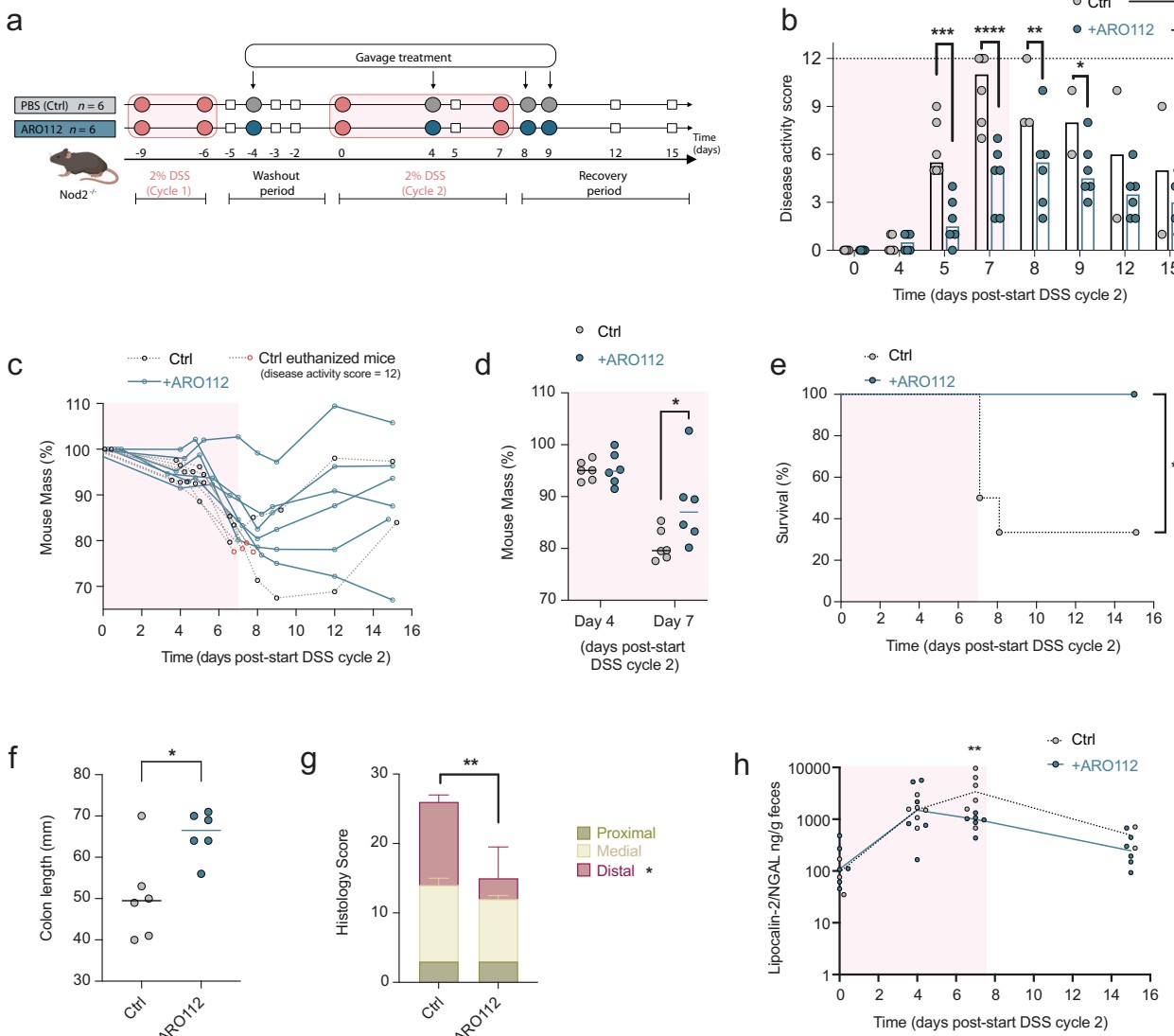

**Fig. 7 | Protective effect of ARO112 against colitis in DSS-treated mice.**
**a** Experimental setup of IBD mouse model Nod2[-/-] mice treated with DSS, and treated with ARO112 or PBS (Ctrl). **b** Disease activity index obtained by the sum of weight loss subscore, stool consistency subscore and bleeding subscore (see Supplementary Data 6; two-sided Mixed-effect analysis with Sidak correction for multiple comparisons. **c** Mouse body weight loss during experiment and **d** at days 4 and 7. Four of six mice from control group (Ctrl) were euthanized after reaching a disease activity index of 12. **e** Survival curve showing percentage of mice that were euthanized due to reaching the experimental endpoint (disease activity index of 12; Log-rank (Mantel-Cox) test; *$p < 0.05$). **f** Colon length of mice euthanized upon

reaching the experimental endpoint or sacrificed at the end of the experiment (two-tailed Mann-Whitney test; *$p < 0.05$). **g** Histological score obtained from the colon of animals (two-sided Mixed-effect analysis with Sidak correction for multiple comparisons; **$p < 0.01$). **h** Fecal lipocalin-2/NGAL levels measured throughout the experiment (two-sided Mixed-effect analysis with Sidak correction for multiple comparisons; **$p < 0.01$). This experiment was performed in 6 mice per group in two independent experiments ($n = 3$ per group per experiment). Panel a was adapted from a figure created in BioRender. Oliveira, R. (2025) https://BioRender.com/i1ikv6t.

the experiment, treatment with ARO112 provided some protection from body weight loss, as seen at day 7, last day of the second DSS cycle, in which ARO112-treated mice lost significantly less mass after two doses of probiotic treatment than mice in the control group ($p = 0.0182$, Fig. 7c, d).

Two-thirds of mice (4 of 6) from the control group developed both severe diarrhea and hematochezia and lost more than 20% of body weight by day 7 or 8 (Fig. 7b, c), and, therefore, did not survive until the end of the experiment (Fig. 7e, log-rank (Mantel-Cox) test, $p = 0.0178$). Remarkably, no mouse treated with ARO112 developed gross rectal bleeding, and all 6 animals from this group survived the experiment and showed recovery of body weight and/or stool consistency (Fig. 7b, c, e Supplementary Data 6). Mice that reached the

maximum DAI were dissected at that timepoint (three mice from the control group at day 7, another at day 8) and all surviving mice (2 from control group and 6 treated with ARO112) were dissected at day 15 of the experiment. Colon length measurement showed that mice of the control group had significantly shorter colons than mice treated with ARO112 (Fig. 7f, two-tailed Mann-Whitney test, $p = 0.0303$), indicating development of colitis and intestinal inflammation in the control group. Similarly, histopathology analysis of medial and distal regions of the colon shows the significantly higher pathological scores in mice from the control group (Fig. 7g, two-sided mixed-effect analysis with Sidak's correction for multiple comparisons, $p = 0.0090$), confirming the protection from colitis conferred by ARO112 probiotic treatment. Protection from intestinal inflammation, is also supported by the fecal

lipocalin-2 measurements (Fig. 7h, two-sided mixed-effect analysis with Sidak's correction for multiple comparisons, $p = 0.0022$), which showed an arrest of intestinal inflammation development between day 4 and day 7 in the mice treated with ARO112.

When analyzing the microbiota composition of Nod2$^{-/-}$ mice subjected to DSS treatment, we noticed that the pro-inflammatory treatment did not result in a dysbiotic state, as we had seen before with the antibiotic treatment, since no significant decrease in richness and diversity levels was observed throughout the experiment (Supplementary Fig. 6a, b). Nevertheless, despite the mild effect of DSS on the overall microbiota composition (Supplementary Fig. 6a–g), we still noticed a decreased number of taxa depleted from untreated levels at the end of the experiment in mice treated with ARO112, when compared to the surviving two animals that did not receive treatment. These included potential butyrate producers (Supplementary Fig. 6c,). We also assessed the fecal levels of butyrate at day 7 and found an increase of butyrate in mice treated with ARO112 close to significance ($p = 0.0625$), which is the timepoint where treated and untreated mice show more significant differences in disease activity as well (Supplementary Fig. 6h, Wilcoxon test). These results show the potential therapeutic effect of ARO112 probiotic treatment against pathologies associated with intestinal inflammatory conditions triggered by DSS in IBD-prone mice, protecting from the development of intestinal inflammation and pathologies, even in the absence of an infectious agent. Additionally, as we observed no major changes in microbiota composition in this model, these results provide evidence that ARO112 therapy can be protective against inflammatory-related pathologies even in the absence of a major dysbiotic event. Thus, overall, this study shows the relevance of ARO112 in promoting recovery from infections, microbiota dysbiotic states, and inflammatory-associated pathologies.

By compiling the different measurements done in these Nod2$^{-/-}$ mouse models, we can assess the implication of probiotic treatments or lack thereof towards disease or healthy markers. Using early (day 5) and late (day 20) infection status, early (day 5) and late (day 20) Lipocalin-2 levels, and late fecal butyrate levels (day 20), and correlating them with the different probiotic treatments (ARO112 and EcN) and control treatment (PBS), we see that only ARO112 treatment is positively associated with late increased butyrate (healthy marker), and negatively associated with early and late increased Lipocalin-2 levels and late infection status (disease markers) (Supplementary Fig. 7a). Butyrate negatively correlates with infection status and Lipocalin-2 levels. Control treatment shows no association with any of these factors, while EcN treatment, in this specific model, appears to correlate positively with late increased Lipocalin-2 levels and infection (Supplementary Fig. 7a).

In the colitis model, a correlation analysis between treatments (PBS or ARO112) and average disease activity index, colon shortening (using the inverse of the colon length at time of sacrifice−1/colon length), and fecal lipocalin-2 measured at the last day of DSS treatment, show that ARO112 treatment correlates negatively with all three disease markers, while PBS treatment associates positively with disease indicators (Supplementary Fig. 7b, two-tailed Kendall's τ rank correlation, p < 0.05).

These results indicate that the only condition tested that consistently leads to healthier outcomes is the treatment with ARO112, in both infection and colitis model.

## Discussion

In this study, we show that *Klebsiella* sp. strain ARO112 is able to displace infection by the pathobiont AIEC in an IBD infection model and attenuate disease in an IBD model with induced colitis. The ability of ARO112 for infection displacement is beyond its direct competitive interaction with AIEC, as it also facilitates microbiota recovery after treatment with antibiotics. Importantly, colonization with this strain leads to higher butyrate intestinal levels, likely due to ARO112

promoting specific members, like Lachnospiraceae and other potential butyrate producers. These findings highlight the potential of ARO112 as a new biotherapeutic agent, as it provides functional consequences and benefits both for the host, by preventing mild intestinal inflammation, and its microbiota, by promoting infection clearance and recovery of microbiota diversity. Importantly, even in the absence of infection, ARO112 protected the host from inflammation-related pathologies, including severe diarrhea and rectal bleeding in a model of chronic inflammation and colitis. Moreover, ARO112 translational potential for humans is supported by analysis of the microbiota in the HMP2 cohort, which shows that participants carrying non-*pneumoniae Klebsiella* species exhibit lower inflammation levels whenever these species were detected in their stool samples. This association indicates that specific strains of these bacterial species might become important players at preventing inflammation in humans, and highlights the need for future human studies to investigate their role in host-microbe interactions.

Other recent studies also add encouraging evidence that other microbiota, non-*pneumoniae Klebsiella* species and strains, might contribute significantly to protection against pathogenic Enterobacteriaceae infections in humans, with support from studies using animal models[37–40]. Specifically, several *K. oxytoca* strains present in children's gut microbiota were found to be protective against invading *K. pneumoniae*[38] and *S.* Typhimurium[40]. Additionally, several bacterial sequences matching non-*pneumoniae Klebsiella* species (related to *K. michiganensis*, *K. oxytoca*, and ARO112) have been associated with protection against intestinal pathogens in cancer patients following therapies that highly increase the risk of bacteremia[39]. In these studies, similar to what we had already shown in a previous publication where ARO112 protected against a laboratory strain of *E. coli* and *S.* Typhimurium[37], protection from these pathogens by non-*pneumoniae Klebsiella* gut isolates depends on nutrition competition. However, our study shows that, at least for ARO112, the protection capabilities of these bacteria go beyond colonization resistance by nutrition competition.

Here, we further expanded the inhibitory capacity of ARO112 showing that it inhibits colonization of multiple Enterobacteriaceae, including MRE strains and non-Enterobacteriaceae species like *V. cholerae*. Importantly, our results showed that the efficacy of the inhibition can vary, and colonization clearance of AIEC is possible only in the presence of other microbiota members. These results indicate that direct nutrition competition and inhibitory mechanisms potentially involved in the interactions between these pairs of bacteria are not sufficient to fully explain our striking result that ARO112 can clear AIEC in the IBD mouse model, and support the importance of the role of ARO112 in promoting microbiota recovery for pathogen clearance. Displacement based on direct interactions, although useful in many situations, is likely to be case specific as they depend on genomic profiles of the protective-pathogen pair[76,77]. On the other hand, the ability of ARO112 to promote recovery of butyrate production by the microbiota, indicates a potential broader therapeutic potential for this specific strain. ARO112 could be effective against a wider range of pathogenic strains and also in other disease states related to dysbiosis not caused by pathogen infections. In accordance with this notion, our results show that ARO112 therapy can decrease intestinal inflammatory pathologies in a chemically-induced model for chronic inflammation and epithelial damage. These results clearly support the potential broad applications of ARO112 therapy in patients suffering from infections, intestinal inflammation, or dysbiosis, but also of a combination of these symptoms.

ARO112's ability to promote recovery of butyrate producers, namely of members of the Lachnospiraceae family, and the increase of butyrate levels, can have promising beneficial results. Butyrate is known to be important for gut health due to its role in promoting epithelial metabolism and immune regulation and having anti-

inflammatory properties[70,78,79]. Butyrate has also been shown to provide colonization resistance to *Salmonella* infections[80,81] and to MDR Enterobacteriaceae[82,83] by direct inhibition. Moreover, other studies have shown the importance of Lachnospiraceae in colonization resistance against multiple pathogens, such as vancomycin-resistant enterococci (VRE)[84], *Clostridium difficile*[85], *Salmonella* Typhimurium[77], and *Listeria monocytogenes*[86]. We have also previously observed that the presence of microbiota members from the *Lactobacillus* genus could restrict MRE colonization by promoting the expansion of Clostridiales, with an increase in butyrate levels observed in both mice and antibiotic-treated leukemia patients[83]. However, in that study, even though addition of *Lactobacillus* could decrease MRE gut invasion by two orders of magnitude, complete clearance was never obtained. Additionally, cocktails containing *Lactobacillus* are often used as probiotics but can delay natural microbiota compositional recovery in healthy individuals[33]. In contrast, the effect obtained by ARO112 treatment in the IBD mouse model resulted in clearance of the pathobiont AIEC with a success rate of more than 80%, alongside an accelerated microbiota recovery.

Other studies also support the notion that many mechanisms of colonization resistance require positive interactions among unrelated microbiota members to promote colonization resistance and pathogen clearance[83,84]. Here, we describe an innovative ARO112 therapeutic effect that goes beyond infection clearance, showing that in an IBD model, ARO112 also promotes the recovery of native butyrate producers depleted by antibiotic treatment and can protect from inflammatory-related pathologies during induced colitis. Therefore, the therapeutic effect of ARO112 is multifactorial, affecting all three main hallmarks of intestinal inflammatory disorders: susceptibility to infection, dysbiosis, and gut inflammation. The relationship between butyrate and inflammation is important, with studies that show the anti-inflammatory properties of this metabolite[70], and other studies that demonstrate the negative impact of inflammation on butyrate producers[87]. This loop-like interaction complicates the manipulation of this metabolite with important therapeutic potential, which further gives significance to ARO112 therapy that results in both the increase of butyrate producers and butyrate, with consequent anti-inflammatory properties, and the prevention of inflammation, subsequently fostering a favorable environment for the recovery of butyrate producers when needed. This interesting interaction between ARO112 and families of butyrate producers like Lachnospiraceae and Ruminococcaceae should be studied further to dissect the functions and mechanisms driving these interactions with such therapeutic potential.

Currently available probiotics therapies often show low efficacy largely due to common caveats: 1) unable to colonize long enough to actuate, 2) cause delay in natural microbiota recovery from dysbiosis and 3) contribute to the increase of antibiotic resistance transfer[33,34,88,89]. However, ARO112 strain does not display any of these caveats in the IBD murine model studied here. ARO112 not only does not delay microbiota recovery in the dysbiotic model (antibiotic-treated mice), but actually promotes recovery while also having the advantage of being consequently cleared from the gut. This is in striking contrast with what happens with other probiotic regimens[33], similar to our results with the probiotic EcN in this specific model, which delayed microbiota recovery, and prolonged AIEC colonization. We tested EcN due to literature suggesting its potential against AIEC[90], supported by both in vitro[91] and in vivo[64] experimentation. A closer look, however, demonstrates that our results do not contradict previous results, since the closest model to ours that has been tested, to our knowledge, was a murine model of DSS-treated wildtype mice in which AIEC infection was tested with or without EcN[92]. In that analogous mouse model, EcN had only a mild inhibitory effect on AIEC colonization (around 10-fold), and similarly to what we observed here,

no AIEC clearance by EcN was observed. Regardless, these results with AIEC, EcN has been shown to be an effective probiotic option in other models and targeting other pathogens and diseases, establishing itself as an advantageous probiotic therapy in use, highlighting the relevance of the comparison between EcN and ARO112, where, at least under the conditions tested here, ARO112 led to a better outcome.

Despite the encouraging support for the role of these protective non-*pneumoniae Klebsiella* in our IBD models of infection and colitis and previous studies on colonization resistance, the fact that these bacteria are also Enterobacteriaceae could raise concerns for clinical applications. Enterobacteriaceae pathogens are the major cause of hospital infections in Europe[93]. Remarkably, genome analysis of ARO112 in comparison with other Enterobacteriaceae revealed that ARO112 has less predicted pathogenic traits. Moreover, the low pathogenic potential based on predicted functions was also supported by our phenotypic tests for traits associated with nosocomial infections. Importantly, our results with the phenotypic tests for pathogenic traits also show that some phenotypes are context-dependent, highlighting the importance of testing cells cultured in different media and, if possible, cells extracted from colonization experiments, before asserting the safety of a specific strain for therapeutic purposes. Notably, ARO112 demonstrated to have a lower pathogenic potential in comparison with the clinical isolate Kp1012, and similar or lower than our control probiotic strain, EcN, which has previously been reported[35,36] to expresses several traits that are commonly associated with pathogenesis. Nevertheless, as EcN is a probiotic used for decades, demonstrating an acceptable degree of safeness, these results are promising for ARO112 therapeutic future. Additionally, several of our experiments show that ARO112 colonization does not appear to induce any immune-related reaction by the host, such as body weight and temperature changes, blood glucose levels changes, or intestinal inflammation development, nor does it translocate to extra-intestinal organs.

In hospitals, the biggest concern related to Enterobacteriaceae infections is the high prevalence of MDR[94], which results in critical clinical situations due to a lack of available alternative treatments. We showed that ARO112 is not MDR but also presents a low capacity for acquiring and retaining plasmids carrying antibiotic resistance. Interestingly, genome analysis of ARO112 also revealed no evidence for genes predicted to be involved in conjugation, which is a major mechanism for plasmid acquisition in Enterobacteriaceae[95], and the absence of these genes in this bacterium might explain its low capacity in plasmid acquisition. These results pose an interesting starting point for further studies to identify the mechanistic basis of this low efficiency for plasmid acquisition, as well as its dissemination in other species and strains of Enterobacteriaceae.

Despite the fact that non-*pneumoniae Klebsiella* species, like ARO112, are at very low abundances in homeostatic conditions[37], oftentimes even below detection levels for taxonomic studies, their effects can still be noticeable. Being able to assess a relationship between carriers of non-*pneumoniae Klebsiella* species and higher levels of potential butyrate producers in humans is very promising, as it potentially validates our findings in animal models that ARO112 can promote higher levels of these bacterial families in humans as well. More importantly, the data showing that within subjects carrying non-*pneumoniae Klebsiella* species, samples with detectable levels of these bacteria have significantly lower levels of inflammation than samples in which these species cannot be detected is very encouraging. Our results are limited by the available human data and thus motivate the need for human studies designed to investigate the effect of ARO112 and related strains in disease status of patients with intestinal inflammation. In the future, designing proper human studies with protocols optimized for the detection and accurate identification of low abundance non-*pneumoniae Klebsiella* species similar to ARO112, will expectedly increase the number of human samples with detectable

non-*pneumoniae Klebsiella* species and will allow for a better understanding of the role of these bacteria in the human microbiota.

Having a safe protective microbiota member that can be used to promote a faster recovery of an imbalanced microbiota and displace pathogens or pathobionts, while preventing inflammatory flares in a disease context is of great therapeutic importance. This safe and effective approach opens new avenues for probiotic research and therapeutic development, offering potential solutions for a range of other clinically-relevant disorders. Further exploration of ARO112's protective mechanisms under different contexts will advance our understanding and enable the identification of additional safe therapeutic microbiota members or consortia for clinical use.

## Methods

### Ethical statement
This research project was ethically reviewed and approved by the Ethics Committee of GIMM (license reference: A003.2022), and by the Portuguese National Entity that regulates the use of laboratory animals (DGAV - Direção Geral de Alimentação e Veterinária (license reference: 015190). All experiments conducted on animals followed the Portuguese (Decreto-Lei nº 113/2013) and European (Directive 2010/63/EU) legislations, concerning housing, husbandry and animal welfare.

### Bacterial strains, plasmids and culture conditions
See Supplementary Data 1 for all species, strains, and plasmids. Unless otherwise specified bacteria were cultured in Lysogenic Broth (LB, here refereed as rich medium) or M9 minimal medium (47.7 mM NaHPO$_4$-7H$_2$O, 22 mM KH$_2$PO$_4$, 8.6 mM NaCl, 18.7 mM NH$_4$Cl, 2 mM MgSO$_4$, 0.1 mM CaCl$_2$, 1 mM thiamine)[96] with 0.5% glucose at 37 °C with aeration (shaking at 240 rpm or in static cultures, as specified).

### Whole genome sequencing
Ec1898, Kp1012, and Kp834 were isolated from three leukemia patients from the Hospital La Fe in Valencia (Spain). Isolation and phenotypic characterization of the strains was performed using selective media for multidrug-resistant Enterobacteriaceae, matrix-assisted laser desorption/ionization–time of flight and the Vitek system, as previously described[17]. The protocol for sample collection was approved by the Ethics Committee of CEIC Dirección General de Salud Pública y Centro Superior de Investigación en Salud Pública (20130515/08). All patients gave their consent for the collection and use of samples. Participants did not receive any compensation for participating in the study. *Klebsiella oxytoca* MBC022 was isolated by plating on LB agar the feces of a WT specific pathogen-free (SPF) C57BL/6 J mice bred in the animal house facility at GIMM. DNA isolation was performed using a previously described protocol[97] from cells in liquid LB at 37 °C with agitation. DNA library and whole genome sequence was obtained at the GIMM Genomics facility. Paired-end sequencing of each sample was performed using an Illumina MiSeq Benchtop Sequencer (Illumina), which produced datasets of 250 bp read pairs.

All sequences have been deposited in the European Nucleotide Archive (ENA) under the study accession number PRJEB102263.

### Method for phylogenetic tree
Phylogenetic tree was obtained using the whole genomes of strains uploaded to the PATRIC BV-BRC browser software (v3.29.20). Parameters used included 100 single-copy genes, using Multiple Alignment using Fast Fourier (MAFFT) Transform alignment program and RAxML Fast Bootstrapping branch support method (v8.2.11).

### Genome-encoded predicted pathogenic properties and gene products
Genomes of all strains were uploaded to the PATRIC BV-BRC[43] browser software (v3.29.20) and analyzed. For each genome, a table containing all predicted pathogenic properties from nine available databases: 3

for Virulence Factors (Victors, PATRIC_VF, VFDB), 3 for Drug Targets and Transporters (DrugBank, TCDB, TTD), and 3 for antibiotic resistances (PATRIC, CARD, NDARO). The ensemble of these data was downloaded and two merged tables were compiled with the information for all the strains: one displaying the presence or absence of each pathogenic property in each strain, and another including the abundance of each pathogenic property in each strain. Principal Coordinates analyses were performed using either presence/absence of pathogenic properties (Bray-Curtis dissimilarity index for binary/categorical data; one-way PERMANOVA with Bonferroni correction, 999 permutations) or abundance of said pathogenic properties (Euclidean distances for continuous data; one-way PERMANOVA with Bonferroni correction, 999 permutations).

Genome-encoded gene products relating to clinically-relevant categories (conjugation and conjugal proteins, natural competence, multidrug resistance, transposases, bacteriophages and integrases, CRISPR systems) number of hits were obtained (Supplementary Data 3) and normalized in graphpad Prism software: in each category the highest number was listed as 100%, the lowest as 0%, and the other numbers distributed according to the percentage within this interval.

### Venn diagrams construction
We analyzed the pathogenic properties that are ubiquitous (present in all five strains) or non-ubiquitous (not present in all 5 strains) of each group (Supplementary Data 2). In the non-*pneumoniae Klebsiella* (npK) strains, there are 64 absent properties when compared to *K. pneumoniae* (Kp) strains, indicating an overall lower pathogenic potential predicted for the npK group. The difference in absent predicted pathogenic properties of the npK strains was even higher when compared with *E. coli* (Ec) strains (403), which mainly included genes encoding for transporters, membrane proteins, and genes potentially conferring resistance to antibiotics (Fig. 1b, Supplementary Fig. 1, Supplementary Data 2). There are other properties absent in the npK strains, but present in the Ec strains that are notable, such as genes related to lipopolysaccharide (LPS) biosynthesis and assembly, flagellum, type IV pilus, toxins, ethanolamine utilization, and hemolysin; all of which having been associated with virulence and pathogenesis (Supplementary Data 2)[98–101]. We investigated the gene products and predicted genome-encoded pathogenic traits in common within each clade, i.e., shared by all five strains tested per clade, similarly to the aforementioned analysis for pathogenic properties. A comparison between gene products, which consider the whole genome (Supplementary Data 3), and the predicted genome-encoded pathogenic properties (Supplementary Data 2), shows that the three clades share a higher percentage of predicted pathogenic traits (47.36%) than gene products (14.23%; Supplementary Fig. 1a).

### Antibiotic resistance profile
Cultures were grown in either rich or minimal medium for 24 h at 37 °C with shaking. Cultures were washed once in sterile PBS and adjusted to a final OD$_{600}$ of 0.05 in 150µl/well in the appropriate medium with different concentrations of antibiotics (as specified). These cultures were then grown for 24 h in a 96-well plate in a plate shaker, after which OD$_{600}$ measurements were performed in a plate reader Multiskan Sky. Strains were considered resistant if in the presence of antibiotic could reach at least 80% of the growth obtained in the absence of antibiotic. If growth in the presence of antibiotic was more than 40% and less than 80% of the control growth, the strain was considered intermediate for resistance, and sensitive if grown less than 40% of control growth. Resistance or intermediate resistance to one antibiotic of any class is considered resistance or intermediate resistance to that class of antibiotics. Sensitivity to a class of antibiotics means that bacteria were sensitive to all antibiotics tested of that class. A strain was considered multidrug resistant (MDR) if it was non-susceptible (sensitive or

intermediate) to at least 1 antimicrobial agent in 3 or more antimicrobial classes, in both tested media, as defined before[17].

## Biofilm formation

Biofilm formation assays were performed by adapting a previously published method (Reisner, 2006; O'Toole, 2011). In summary, bacterial cultures were grown for 24 h in either rich or minimal media at 37 °C with shacking. The following day, each culture was washed once in sterile PBS, and final $OD_{600}$ adjusted to 0.05 in the appropriate medium and 200 µl of culture were dispensed per well in 48-well plates. Plates were centrifuged for 5 min at 2057 xg at room temperature, after which plates were incubated at 37 °C for 90 min for adhesion. After adhesion, supernatants were carefully removed and discarded, wells were washed once with 200 µl of sterile PBS, and 500 µl of sterile fresh appropriate medium were carefully dispensed per well. Plates were incubated at 37 °C for 24 h, after which attached biofilms (biomass) was quantified with Crystal Violet (CV). For that supernatants were removed and discarded and well bottoms were carefully washed with 500 µl of sterile PBS. Each well was carefully stained with 0.1% CV solution and plates were incubated at room temperature for 20 min protected from light. After incubation, each well was washed with PBS and plates were incubated open and inverted in a paper towel for 30 min protected from light, to dry. Each well was then de-stained with 200 µl of a 33% glacial acetic acid solution and incubated for 15 min at room temperature protected from light. Supernatants were removed from each well into a new plate and read for $OD_{580}$ in a Multiskan Sky plate reader (peak for CV staining), to quantify biofilm biomass.

## Urease activity

Cultures were grown in either rich or minimal medium for 24 h at 37 °C with shaking. Cultures were washed once in sterile PBS and adjusted to an $OD_{600}$ of 1, after which each culture was diluted 20x in urea medium (3 g/L L-tryptophan, 5 g/L NaCl, 1 g/L $H_2KO_4P$, 1 g/L $H_2K_2O_4P$, 20 g/L urea, 0,012 g/L phenol red) and incubated for 24 h at 37 °C with shaking. Urea degradation leads to the medium color to change from yellow/orange to pink/red. 100 µl of each culture were then transferred into a 96-wellplate and measured at $OD_{560}$ in a Multiskan Sky plate reader to quantify urease activity. For urease activity measurement of bacteria extracted from fecal samples of monocolonized mice, cell suspensions of $2.5 \times 10^7$ cells/ml were used, instead of the grown culture.

## Siderophore production

Cultures were grown in either rich or minimal medium for 24 h at 37 °C with shaking. Cultures or fecal samples diluted in PBS were centrifuged to recover supernatants. Culture and fecal supernatants were filtered in 0.22 µm filters. Cell-free supernatants (100 µl) were mixed 1:1 (v:v) with Cas-PIPES solution (100 µl) (Alexander & Zuberer, 1991; Das & Barooah, 2018) in a 96-well plate and incubated at room temperature for 20 min, after which $OD_{630}$ was measured in Multiskan Sky plate reader to calculate percentage of siderophore production, using sterile media or PBS as reference.

$$\%siderophores = \frac{(ref - sample)*100}{ref}$$

## Resistance to pooled human serum

Bacterial susceptibility to human serum was tested as described[102] with modifications. Bacterial strains were grown in either rich or minimal medium for 24 h at 37 °C with shaking, or extracted directly from fecal pellets. Cultures were washed once in sterile PBS and adjusted to $2 \times 10^7$ cells/ml, after which 25 µl of cell suspension in PBS were added to 75 µl of pooled human serum solution (P30-2401, PAN-Biotech). Cultures were plated in LB agar plates for CFUs at 0 h and after 3 h of static incubation at 37 °C, to assess killing, resistance, or proliferation of bacterial cells in the presence of human serum.

## Plasmid conjugation

Cultures were grown in rich medium for 24 h at 37 °C with shaking. Cultures were washed once in sterile PBS. Donor (MG1655 RP4, Kgri RP4) and recipient (ARO112, EcN, Kp1012) strains were thoroughly mixed at a 1:1 ratio (total of $10^8$ cells), centrifuged for 30 sec at 20,000 xg and resuspended in 20 µl of PBS. Two drops of 10 µl each were placed on an agar plate, air-dried, and incubated at 37 °C for 24 h. After incubation, both drops were removed with a sterile loop and resuspended in 1 ml of sterile PBS. Serial dilutions were performed and plated to count CFUs for recipient strain with (selective plating with plasmid-borne antibiotic resistance) and without (non-selective or selective plating for recipient's antibiotic resistance plating) tested plasmid. Conjugative plasmid (RP4) harbors resistances to tetracycline (10 µg/ml) and kanamycin (50 µg/ml).

## Plasmid retention

Strains harboring tested plasmids (RP4 acquired through conjugation from MG1655 RP4 donor strain; pMP7605 acquired through electroporation; Kanamycin resistance acquired through conjugation from clinical isolate Kp824) were grown in either rich or minimal medium with antibiotics (for plasmid-borne antibiotic resistances; kanamycin (50 µg/ml) and tetracycline (10 µg/ml) for RP4; gentamicin (30 µg/ml) for pMP7605; kanamycin (50 µg/ml) for transfer from clinical isolate) for 24 h at 37 °C with shaking. Cultures were washed once in sterile PBS and adjusted to a final $OD_{600}$ of 0.05 in 150 µl in the appropriate medium without antibiotics, and serial dilutions were plated in selective and non-selective LB agar plates, to assess CFUs with (plasmid retention) and without (plasmid loss) antibiotic resistance. Cultures were incubated at 37 °C with shaking for 24 h. Every 24 h, cultures were diluted 1:100 in fresh appropriate medium and re-incubated. After 5 days of passages, serial dilutions of cultures were plated as mentioned before.

## Animal experimentation

All mice (*Mus musculus*) used in this study were supplied by the Rodent Facility at GIMM and were given *ad libitum* access to food (Rat and Mouse No.3 Breeding – Special Diets Services, product number 801030) and water. Mice were kept at 20 °C–24 °C and 40–60% humidity with a 12 h light-dark cycle.

C57BL/6 J mice, either WT or Nod2$^{-/-}$, were used at 6–24 weeks of age and were randomly assigned to experimental and control groups and kept singly caged throughout the experiment. C57BL/6 J mice were used for all experiments. None of the animal experiments were performed blinded. Sample size was chosen according to institutional directives and in accordance with the guiding principles underpinning humane use of animals in research. No statistical analyses were performed to predetermine the sample sizes. All of the experiments were performed at least twice, except when stated otherwise.

## Bacterial isolation from fecal samples

WT C57BL/6 J germ-free mice were orally gavaged with 100 µL containing ~$10^8$ CFUs of ARO112, EcN, or Kp1012 strains bearing the non-self-transmissible plasmid pMP7605. Five days later, fecal pellets were collected for live gut bacteria extraction, as previously described[53]. Briefly, pellets were weighed and 500 µL of sterile PBS were added before mechanical disruption with a motorized pellet pestle, after which another 500 µL of sterile PBS were added. Samples were subjected to four iterations of vortex mixing for 15 s, centrifugation at 106 xg for 30 s at room temperature and recovery of 750 µL of the debris-free cell-containing supernatants into a new tube, replacing that volume with sterile PBS before the next iteration. After recovery of 3 ml per sample,

isolated cells were centrifuged at 4000 x g for 5 min at room temperature, discarding the supernatant and resuspending the cells fractions in 1 ml of sterile PBS supplemented with glycerol and cysteine at final concentrations of 20% and 0.1%, respectively. Samples were frozen and stored at −80 °C, to measure phenotypes (urease activity, human serum resistance, plasmid retention) as described above. Fecal supernatants were used to assess fecal Lipocalin-2 levels and siderophores.

## Nod2$^{-/-}$ and WT SPF mice experiments

C57BL/6 J Nod2$^{-/-}$ and WT mice bred under specific pathogen-free conditions in the animal house facility at the GIMM were kept in sterile ISOcages (Tecniplast) at our specific opportunistic pathogen-free facility. At the start of the experiment, mice were individually housed and were orally gavaged with gentamicin (3 mg/kg/d) and vancomycin (40 mg/kg/d) once daily for three consecutive days[62]. Two days after the last gavage with antibiotics, mice were orally gavaged with 100 μL containing ~$10^8$ CFUs of AIEC strain (LF82-strepR), and the following day were orally gavaged either with 100 μL of PBS (control), or 100 μL containing ~$10^8$ CFUs of ARO112 or EcN strains. All bacteria were grown overnight in LB medium and the day of the gavage, bacteria were diluted 1:100 and grown to $OD_{600} = 2$, then medium was removed by centrifugation and cells were resuspended in PBS. Fecal samples were collected at the timepoints indicated in the corresponding graph and plated in selective medium to assess colonization levels of AIEC. Additionally, fecal samples were also used to measure levels of Lcn2 (Lipocalin-2/NGAL ELISA) and to determine the intestinal microbiota composition and levels of SCFA.

C57BL/6 J Nod2$^{-/-}$ litter-paired mice at 6-24 weeks of age were separated into two groups (control and ARO112-treated), were kept individually in sterile ISOcages (Tecniplast) at our specific opportunistic pathogen-free facility and treated with 2% DSS in the drinking water (DSS supplemented-water was replaced every other day), in two cycles, the first for 4 consecutive days and the second for 7 consecutive days, with two washout periods of 4 and 7 days, respectively, with non-supplemented water.

## WT and Nod2$^{-/-}$ gnotobiotic mice colonization clearance experiments

C57BL/6 J WT or Nod2$^{-/-}$ germ-free mice were bred and raised in GIMM's Gnotobiology Unit in axenic isolators (La Calhene/ORM) and were later transferred into sterile ISOcages (Tecniplast). Animals were individually housed and orally gavaged with 100 μL containing ~$10^8$ CFUs of AIEC (LF82-strepR), Kp1012, *V. cholerae*, or Ec1898 strains, or ~$10^8$ CFUs of ARO112 or ~$10^4$ CFUs of *S*. Typhimurium (mono-colonization experiments). The following day, mice were gavaged with 100 μL containing ~$10^8$ CFUs of either ARO112 or EcN strains, or 100 μl of PBS (except for mono-colonization experiments). Fecal samples were collected at the timepoints indicated in the corresponding graphs and plated in selective media to assess colonization levels of AIEC, Kp1012, *V. cholerae*, EC1898, or *S*. Typhimurium strains.

## 16S rRNA sequencing

Fecal samples from SPF mice experiments stored at −80 °C were processed for DNA extraction as previously described[103]. In brief, DNA was extracted using a combination of the QIAamp Fast DNA Stool Mini Kit (Qiagen) according to the manufacturer's instructions, mechanical disruption using a motorized pellet pestle and elution to a final volume of 100 μL in ATE buffer. For each sample, the 16S rRNA gene was amplified using the 515 F/806 R (V4 regions) primer pairs recommended by the Earth Microbiome Project under the following PCR cycling conditions: 94 °C for 3 min; 35 cycles of 94 °C for 60 s, 50 °C for 60 s and 72 °C for 105 s; and extension at 72 °C for 10min[104,105]. After library preparation, 2 × 250 bp sequencing was performed at the GIMM Genomics Unit using an Illumina MiSeq Benchtop Sequencer.

## Fecal microbiota analysis

Mothur v.1.32.1 was used to process sequences as previously described[103], with some modifications. Sequences were converted to FASTA format. Sequences that were shorter than 220 bp containing homopolymers of longer than 8 bp or undetermined bases with no exact match with the forward and reverse primers, and barcodes that did not complement each other or that did not align with the appropriate 16S rRNA variable region, were not included in the analysis. A quality score above 30 (range, 0 to 40, where 0 represents an ambiguous base) was used to process sequences, which were trimmed using a sliding-window technique over a 50 bp window. Sequences were trimmed from the 3' end until the quality score criterion was met, and were merged after that. Between 20,000 and 50,000 sequences were obtained per sample. 16S rRNA gene sequences were aligned using SILVA template reference alignment[106]. ChimeraSlayer[107] was used to remove potential chimeric sequences. Sequences with distance-based similarity of greater than 97% were joined into the same OTU using the average-neighbor algorithm. All of the samples were rarefied to the same number of sequences (10,000) for diversity analyses. Samples below 10,000 reads were removed from the analysis. A Bayesian classifier algorithm with a 60% bootstrap cut-off was used for each sequence[108]; sequences were assigned to the genus level where possible or otherwise to the closest genus-level classification. All OTUs were used, and taxa plotted are the result of the merge of OTUs with the same classification, being collapsed to 37 different taxa, with lower represented OTUs/taxa being summed and plotted as "other". Principal Coordinate Analyses were performed using the 37 taxa or a subset of 8 taxa of potential butyrate producers depleted by antibiotics (*Alistipes*, *Syntrophococcus*, *Ruminococcus*, *Butyricicoccus*, *Dorea*, *Oscillibacter*, unclassified Ruminococcacea, unclassified Lachnospiraceae), representing the Bray-Curtis dissimilarity index (one-way PERMANOVA with Bonferroni correction and 999 permutations).

All sequences have been deposited in ENA under the study accession number PRJEB102263.

## Lipocalin-2/NGAL enzyme-linked immunoassay (ELISA)

Fecal samples were weighted and resuspended in 1 ml sterile PBS and mechanically homogenized with a motorized pellet pestle. Samples were centrifuged at 4 °C for 15 min at 18,000 x g. Supernatants were recovered and filtered in 0.22 μm filters. ELISA for Lipocalin-2/NGAL (Human Lipocalin-2/NGAL DuoSet ELISA) were performed according to manufacture instructions, except for the volumes used, which were adapted for 384-well plates, instead of 96-well plates. The following volumes were used instead: reagent diluent (40 μl/well), sample and standards (20 μl/well), Detection Antibody (20 μl/well), Streptavidin-HRP (20 μl/well), Substrate solution (20 μl/well), Stop solution (20 μl/well).

## SCFA quantification

1H-NMR (proton nuclear magnetic resonance) analysis was performed to determine the abundance of SCFA in the fecal samples collected from Nod2$^{-/-}$ mice treated with ARO112 and EcN, or untreated (control), as previously described[109]. Fecal samples (1 to 3 fecal pellets) were diluted in 1 mL of PBS and mechanically homogenized. To remove large debris, the samples were pelleted by centrifugation at 18,000 g for 15 min at 4 °C. Supernatants was collected and filtered through a 0.22 mm filter (Milipore), followed by another filtration step with 3 KDa filters (Vivaspin 500) using centrifugation at 15,000 x g and 4 °C for 3 h (or until 150 μL of filtrate was obtained). Filtered samples were stored at −80 °C until spectra were acquired. For spectrum acquisition, samples were thawed at room temperature and 150 μL of sample mixed with 60 μL of 350 mM phosphate buffer (pH 7.09 with 2% NaN3, 10 μL of a 0.05% (w/v) 3-(Trimethylsilyl)propionic-2,2,3,3-d4 (TSP-d4, Sigma-Aldrich) solution, and 380 μL of deuterated water - $D_2O$) (for a total volume of 600 μL). This mixture was transferred to a 5 mm glass NMR

tube. All solutions were prepared with $D_2O$. Samples were homogenized by inversion and the spectra were acquired after pH measurement. Acquisitions were performed on a Bruker NEO 500 MHz instrument equipped with QXI H-C/N/P 5 mm probe-head with z-gradients. 1H-NMR spectra were acquired using 1D NOESY pulse sequence with pre-saturation (noesypr1d) under the following conditions: 90 degrees pulse for excitation mixing time 100 ms, acquisition time 4 s, and relaxation delay 1 s. All spectra were acquired with 200 scans at 25 °C, with 48,000 data points and 6002 Hz (12 ppm) spectral width (Chenomx acquisition parameters). The recorded 1H-NMR spectra were phase corrected using Bruker TopSpin 4.0.7 and spectra were then analyzed using Chenomx NMR Suite 8.1. Compounds were identified by manually fitting reference peaks to spectra in database Chenomx 500 MHz Version 10. Quantification was based on internal standard peak integration (TSP-d4).

### Human data analysis

Human metadata and taxonomic data on IBD patients (UC and CD) and non-IBD participants from the Human Microbiome Project (HMP2) were downloaded from https://www.ibdmdb.org/results. Taxonomic data and metadata were compiled and samples were sorted: 1) samples collected under antibiotic treatment were removed from the study as it affects microbiota composition; 2) samples lacking data for fecal calprotectin levels were removed from the study as it did not allow for comparison between microbiota composition and inflammation. Of the remaining samples, we sorted participants according to the presence or absence of non-*pneumoniae Klebsiella* species (identified as *K. michiganensis* and *K. oxytoca* in the taxonomic data) in their fecal samples: participants with at least one sample with detectable levels of non-*pneumoniae Klebsiella* were considered "carriers", while participants with no detectable levels of non-*pneumoniae Klebsiella* species in any sample were considered "non-carriers". We have analyzed the abundance of selected potential butyrate producer families (sum of Lachnospiraceae, Oscillispiraceae, and Ruminococcacea), as well as of *E. coli*.

### Software used

Mothur v.1.32.1 was used to process 16S rRNA sequences, while ChimeraSlayer was used to remove potential chimeric sequences. BV-BRC browser software (v3.29.20) was used to build a phylogenetic tree, using Multiple Alignment using Fast Fourier (MAFFT) Transform alignment program and RAxML Fast Bootstrapping branch support method (v8.2.11). Phylogenetic tree (Fig. 1a) was done using iTOL (itol.embl.de). Cluster heatmap (Fig. 1b) was done using SRPlot (https://www.bioinformatics.com.cn), using bidirectional clustering, complete cluster method and Euclidean distances. Networks of interactions (Supplementary Fig. 7) were computed with networkx v3.3 and rendered by matplotlib v3.10.0, using the Jupyter notebook v7.2.2 running Python v3.12.3, including pandas v2.2.3, numpy v1.26.4, and scipy v1.13.1 packages. Principal Coordinate Analyses were obtained with Past4 v02 software (https://folk.universitetetioslo.no/ohammer/past) as explained above. Schematic diagrams were generated in BioRender and figures were assembled and adapted using Adobe Illustrator 2025 v29.8.2.

### Statistics & reproducibility

All statistical analyses presented were done using Graphpad Prism 10 or Past4.02, with statistical tests used being described across the text and in the figure legends, including corrections for multiple comparisons where required. P values lower than 0.05 were considered significant and represented either by different letter across conditions, or different symbols (#, *$p < 0.05$, **$p < 0.01$, ***$p < 0.001$, ****$p < 0.0001$). No statistical methods were used to predetermine sample size. No data were excluded from the analyses. Animals used in in vivo experiments were randomly allocated to the different groups. SCFA quantification

and histopathological scores were blinded to the investigators. Results were reproducible across repetitions.

### Reporting summary

Further information on research design is available in the Nature Portfolio Reporting Summary linked to this article.

## Data availability

The NGS data generated in this study have been deposited in the ENA (European Nucleotide Archive) database under the study accession code PRJEB102263. All other data generated in this study are provided in the Supplementary Information/Supplementary Dataset files.

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

## Acknowledgements

We thank the members of the Bacterial Signaling Laboratory at the Gulbenkian Institute for Molecular Medicine (GIMM), Tanja Dapa, and João Xavier for critically reading the manuscript and Isabel Gordo for valuable feedback. In particular, we thank Joana Amaro for the help processing samples for genomic sequencing and metabolomics. We also thank GIMM's Animal Facility, in particular Joana Bom, the Genomics Facility, namely João Sobral, and the Histopathology Unit, particularly Pedro Faísca, for the help provided during this project's execution. The authors would like to thank Gabriel Nuñez for providing the AIEC strain (LF82), and Roberto Balbontín-Soria for providing the plasmid pMP7605. We would also like to acknowledge Athanasios Typas for the suggestion to test conjugation from a clinical isolate donor. This work was funded by Fundação para a Ciência e Tecnologia (FCT-Portugal) projects [PTDC/BIA-MIC/6990/2020] (KBX) and [SFRH/BPD/116806/2016] (VC), as well as by Municipality of Oeiras Proof-of-Concept InnO-Valley 2022 [IOVPoC-2022-21] (VC) and a European Commission grant [MSCA-IF-2018-843183] (VC). Additionally, this project was also funded by InfectERA-ERANET- through FCT-Portugal grant (InfectERA/0004/2015) (KBX) and Acciones complementarias grant [PCIN-2015-094] from the Spanish Ministerio de Economía y Competividad (CU) and the 7th Research framework program from EU granted to KBX and CU, respectively, and a grant from the Spanish MICINN [PID2023-150086OB-I00] and from Conselleria d'Innovació, Universitats, Ciència i Societat Digital [CIPROM/2021/053], granted to CU. NMR data were acquired at CERMAX, ITQB-NOVA, Oeiras, Portugal with equipment funded by FCT, project AAC 01/SAICT/2016.

## Author contributions

V.C. and R.A.O conceptualized and designed the research studies with advice and support from K.B.X and C.U. R.A.O. executed in vitro experiments with support from V.C. V.C. executed in vivo experiments with AIEC and Kp1012, in SPF wildtype, SPF Nod2$^{-/-}$, and GF wildtype models, with support from R.A.O. M.B.C. performed in vivo experiments with *V. cholerae* and Ec1898 with support from R.A.O. M.F.P. performed the NMR. R.A.O. analyzed microbiome sequencing data and V.C. performed the post-analysis. C.U. and M.G provided the whole-genome sequencing data for human isolates Kp1012, Kp834, Ec1898. V.C. and R.A.O. analyzed and interpreted all the data, under the supervision of K.B.X. V.C., and R.A.O. wrote the first draft of the manuscript with valuable input from K.B.X. and C.U. All authors critically read and approved the manuscript.

## Competing interests

GIMM has filed for a Provisional European Patent Application (EP23184673) regarding this work, in which Vitor Cabral, Rita A. Oliveira, and Karina B. Xavier are the inventors. Carles Ubeda has participated as a consultant of Vedanta Biosciences and The Zambon Group, but there is no direct overlap between the current study and these consulting duties. The rest of the authors do not have any competing interests.
