## [Transparent Peer Review file · Nature Communications]

***Klebsiella* ARO112 promotes microbiota recovery, pathobiont clearance and prevents inflammation in IBD mice**

Corresponding Author: Dr Vitor Cabral

Version 0:

Reviewer comments:

Reviewer #2

(Remarks to the Author)

The authors have addressed my prior comments by including substantial new data that strengthen the manuscript's conclusions. I have no further comments.

Reviewer #3

(Remarks to the Author)

Cabral & Oliveira et al have made a concerted effort to address the comments of all reviewers. They should be commended for their efforts, especially as the points they were asked to address encompassed a wide variety of experimental approaches and required a range of models.

The authors have satisfactorily answered my questions on the safety profile of their probiotic. They also added further details and data to address comments on the inflammation induced within the intestine, and they have also tempered their conclusions in this area. Additional data on the biogeography of the AIEC along the length and span of the intestine is also a good addition. Thanks for addressing this point.

I also appreciate the efforts to address my comments about the efficacy of the probiotics relative to other organisms. My comment was open-ended, and to address it fully would require significant further work, which I agree is beyond the scope of this study.

I'm happy that the authors have adequately addressed all of the comments.
